# On-shelf circulation of warm water toward the Totten Ice Shelf in East Antarctica

Daisuke Hirano [1,2] ✉, Takeshi Tamura [1,2], Kazuya Kusahara [3], Masakazu Fujii[1,2], Kaihe Yamazaki [1,4,5], Yoshihiro Nakayama [6,7], Kazuya Ono [6], Takuya Itaki[8], Yuichi Aoyama[1,2], Daisuke Simizu [1], Kohei Mizobata [9], Kay I. Ohshima [6,7], Yoshifumi Nogi [1,2], Stephen R. Rintoul [10,11,12], Esmee van Wijk [10,12], Jamin S. Greenbaum [13], Donald D. Blankenship [14], Koji Saito[15] & Shigeru Aoki [6,7]

The Totten Glacier in East Antarctica, with an ice volume equivalent to >3.5 m of global sea-level rise, is grounded below sea level and, therefore, vulnerable to ocean forcing. Here, we use bathymetric and oceanographic observations from previously unsampled parts of the Totten continental shelf to reveal on-shelf warm water pathways defined by deep topographic features. Access of warm water to the Totten Ice Shelf (TIS) cavity is facilitated by a deep shelf break, a broad and deep depression on the shelf, a cyclonic circulation that carries warm water to the inner shelf, and deep troughs that provide direct access to the TIS cavity. The temperature of the warmest water reaching the TIS cavity varies by ~0.8 °C on an interannual timescale. Numerical simulations constrained by the updated bathymetry demonstrate that the deep troughs play a critical role in regulating ocean heat transport to the TIS cavity and the subsequent basal melt of the ice shelf.

While the contribution to global sea-level rise from West Antarctica has received the most attention (6.9 ± 0.6 mm for the period 1979–2017, dominated by the Amundsen and Bellingshausen Sea sectors), East Antarctica has also been a major contributor (4.4 ± 0.9 mm over the same period)[1]. The greatest discharge of ice from the East Antarctic Ice Sheet occurs via the Totten Glacier, which drains the Aurora Subglacial Basin in Wilkes Land. The Totten Glacier has an ice volume equivalent to >3.5 m of global sea-level rise[2,3] and is grounded below sea level[4,5], making the Aurora Subglacial Basin vulnerable to oceanic thermal forcing. The Totten Ice Shelf (TIS) experiences an area-averaged basal melt rate of 10.5 ± 0.7 m yr⁻¹, the

highest of all East Antarctic ice shelves, and a basal melt amount of 63.2 ± 4 Gt yr⁻¹ for 2007–2008 (ref. 6).

Ocean heat transport to the ice shelf cavity drives basal melt of ice shelves. High melt rates and ice loss from the West Antarctic Ice Sheet are driven by intrusions of warm Circumpolar Deep Water (CDW)[7,8]. East Antarctic continental shelves are typically cold and fresh[9] because access of warm CDW to the shelf is limited by the presence of the Antarctic Slope Current (ASC) associated with the Antarctic Slope Front (ASF) over the upper continental slope[9,10]. Knowledge of the factors that regulate the transport of ocean heat from offshore to the ice shelf cavity is essential to assess the vulnerability of the Totten

[1]National Institute of Polar Research, Tachikawa, Japan. [2]The Graduate University for Advanced Studies, SOKENDAI, Tachikawa, Japan. [3]Japan Agency for Marine-Earth Science and Technology, Yokohama, Japan. [4]Institute for Marine and Antarctic Studies, University of Tasmania, Hobart, TAS, Australia. [5]The Australian Centre for Excellence in Antarctic Science, University of Tasmania, Hobart, TAS, Australia. [6]Institute of Low Temperature Science, Hokkaido University, Sapporo, Japan. [7]Graduate School of Environmental Science, Hokkaido University, Sapporo, Japan. [8]National Institute of Advanced Industrial Science and Technology, Tsukuba, Japan. [9]Tokyo University of Marine Science and Technology, Tokyo, Japan. [10]CSIRO Environment, Hobart, TAS, Australia. [11]Centre for Southern Hemisphere Oceans Research, Hobart, TAS, Australia. [12]Australian Antarctic Program Partnership, University of Tasmania, Hobart, TAS, Australia. [13]Scripps Institution of Oceanography, University of California, San Diego, CA, USA. [14]Institute for Geophysics, The University of Texas at Austin, Austin, USA. [15]Japan Coast Guard, Hydrographic and Oceanographic Department, Tokyo, Japan. ✉e-mail: hirano.daisuke@nipr.ac.jp

Glacier and its large potential contribution to sea level rise. Substantial gaps remain because most of the continental shelf near the Totten Glacier has not been sampled due to heavy sea-ice cover. Previous observations on the Totten continental shelf are limited to the relatively shallow Dalton Polynya region and a single voyage to the front of the TIS[11,12]. Those observations showed that relatively warm modified CDW (mCDW) is present on the continental shelf and reaches the TIS[11,12], through deep troughs that provide access to the ice shelf cavity[3]. However, from these limited observations, it was not possible to determine the pathways and processes regulating the transport of warm water from the offshore ocean to the TIS cavity, nor was it possible to assess variability in the temperature of mCDW.

Our observations span the Totten embayment, including broad helicopter-based measurements, detailed multibeam bathymetric surveys of the deep troughs that provide warm water access to the cavity, and hydrographic measurements from the unsampled western side of the ice front, where glacial meltwater is speculated to leave the cavity. We collected comprehensive observations of bathymetry and water properties off the Sabrina Coast during three cruises (March 2018, December 2019/March 2020, and March 2022) extending from the upper continental slope/shelf break across the continental shelf to the TIS front (see Methods) and including many regions not previously sampled. Here we combine our bathymetric data with existing measurements to create a more realistic and detailed bathymetric dataset for the region (Methods; Supplementary Fig. 1). We then analyze the distribution and circulation of warm mCDW revealed by the hydrographic data. Multi-year hydrographic observations spanning the period from 2015 to 2022 and moored time series from the deep troughs also allow us to assess temporal variability in the temperature of mCDW reaching the ice shelf cavity. Using the updated bathymetry data, we configure a high-resolution ocean-sea ice-ice shelf model and use the numerical simulation, validated against our ocean observations, to provide additional insight into the pathways of mCDW from the open ocean to the ice shelf cavity. Specifically, we utilize the numerical simulations to estimate ocean heat transport, basal melt, the ocean circulation in the TIS cavity, the connection to the ocean circulation over the continental shelf, and the sensitivity of ocean heat transport to the representation of bathymetry. The observations and model simulations delineate the pathways of warm mCDW from the shelf break to the ice shelf cavity, highlighting the importance of bathymetry and regional circulation in regulating heat transports toward the TIS cavity.

## Results and discussion

### Updated bathymetry off the Sabrina Coast

Due to the presence of heavy sea ice, most of the continental shelf offshore of the Sabrina Coast is difficult to access by ship and the bathymetry is therefore poorly known. Previous ship-based observations were mostly limited to the relatively ice-free Dalton Polynya[13], east of the TIS (Supplementary Fig. 1). We conducted a multibeam bathymetric survey in the uncharted regions over the continental shelf and margin of the Sabrina Coast, from the shelf break to the TIS front, including the sea-ice covered region west of the Dalton Polynya (Fig. 1; Supplementary Fig. 1; see Methods for details).

The updated bathymetry reveals several bathymetric features that facilitate the transport of warm mCDW to the TIS (Fig. 1). The shelf break is relatively deep (bottom depth of 500–600 m) along the full extent of the embayment, allowing warm offshore deep waters to access the continental shelf[14,15]. A broad and deep (600–900 m) depression (the "Sabrina Depression"; Fig. 1) spans most of the continental shelf, allowing warm water that reaches the shelf to spread towards the continental margin. Closer to the coast, the depression shoals to depths of 400–500 m. A previous study[11] had identified the presence of several deep glacial troughs at the TIS front, but it was unclear if the troughs extended north to the continental shelf. The

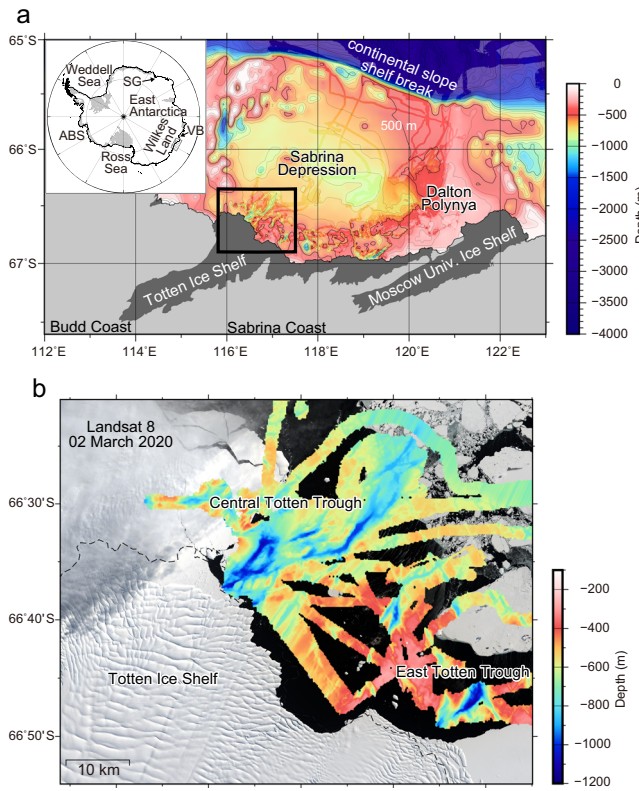

**Fig. 1 | Updated bottom topography off the Sabrina Coast. a** Updated bathymetry off the Sabrina Coast compiled from multibeam sonar data (JARE61 and JARE63) and all available bathymetric data. The distribution of all available multibeam sonar data (see also Supplementary Fig. 1a) is superimposed on the compiled gridded bathymetry (background with transparency, see also Supplementary Fig. 1c). Thick black line denotes the 500-m isobaths. The domain is indicated by the hatched region in the inset map (ABS: Amundsen and Bellingshausen Sea; VB: Vincennes Bay; SG: Shirase Glacier). **b** Detailed spatial structure of bottom topography near the Totten Ice Shelf (TIS), revealed by the multibeam sonar bathymetric survey, superimposed on the satellite image of ice conditions derived from Landsat 8 on 2 March 2020. The domain is indicated by the black bounding box in (**a**).

multibeam sonar data show that these troughs (the "Totten Troughs") connect the Sabrina Depression to the TIS cavity (Fig. 1), providing a pathway for warm water at depth to reach the ice shelf cavity. The updated bathymetry also reveals the detailed structure of the troughs, including several sills that may act to partially block the flow of warm water along the seafloor. The troughs range in depth from 600 to 1200 m and in width from 10 to 20 km. Central Totten Trough (C-TT) at the central TIS front near 116.5°E, extending at least 40 km to the northeast from the TIS front, is the most significant, with a maximum depth of -1100 m and width of -20 km. C-TT connects the TIS cavity and the Sabrina Depression, with the minimum depths along the channel exceeding 600–700 m (Fig. 1b). East Totten Trough (E-TT) at the eastern TIS front near 117.3°E is also deep (maximum depth >1000 m) but appears to be surrounded by shallower topography (<500–600 m depth, Fig. 1b) and, therefore, may be less connected to the deep waters of the Sabrina Depression.

### Pathways of warm water from the shelf break to the TIS front

The distribution of water temperature over the continental slope, shelf break, and inner shelf illustrates how and where mCDW spreads onto and across the shelf (Fig. 2). Over the upper continental slope, distinct subsurface warm cores (0.8–1.3 °C) are observed at 400–500 m depth (Fig. 2a), which are associated with semi-permanent cyclonic eddies[16]

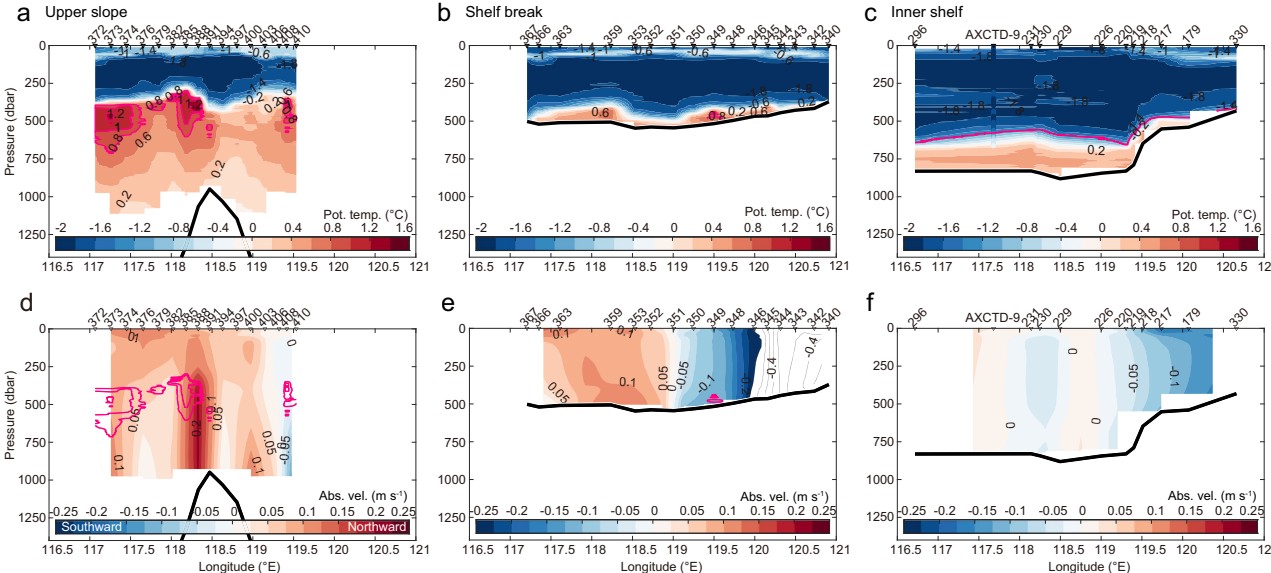

**Fig. 2 | Continental slope-to-shelf water mass distribution and circulation.** Vertical sections of potential temperature (°C) and absolute ocean velocity (m s$^{-1}$) along the (**a**, **d**) upper continental slope, (**b**, **e**) shelf break, and (**c**, **f**) inner shelf sections. The locations of the sections are indicated by the green lines in Fig. 3a. Warm cores with potential temperature >0.8 °C are highlighted in pink contours. In (**e**), absolute ocean velocity values larger than the range covered by the colormap scale are displayed with contours only. In (**c**), in situ temperature is shown for AXCTD cast (AXCTD-9 between XCTD stations 296 and 231) for reference because salinity data were not accurately obtained. In addition, the potential density surface of $\sigma_\theta = 27.7$ kg m$^{-3}$, defined as the upper boundary of mCDW[12], is indicated by the pink line. Each panel shows the bottom topography at each station derived from the multibeam sonar bathymetry data as bold black lines.

(Supplementary Fig. 2) as discussed in previous work[15,17]. Although the upper slope section east of 119°E (Fig. 2a, d) is somewhat distant from the shelf break, the warm water cores over the slope are shallower than the shelf break (500–600 m deep) and can be traced across the shelf break and onto the continental shelf, where mCDW is widespread near the seafloor, with maximum temperatures of ~0.8 °C (Fig. 2).

The water column on the shelf exhibits a two-layer structure, with Winter Water (WW, cold, fresh, and high dissolved oxygen (hereafter DO)) overlying mCDW (warm, saline, and low DO) near the seafloor. These contrasting water masses are separated at ~500–600 m depth by a sharp interface in temperature and salinity (hence density, noting that seawater density is determined mainly by salinity in the Antarctic coastal region) (Fig. 2c; Supplementary Fig. 3). A zonal temperature section crossing the deep basin on the shelf shows that relatively warm (0–0.5 °C) mCDW occupies the bottom layer of the entire Sabrina Depression (Figs. 2c, 3a; Supplementary Fig. 3). Absolute ocean velocity (estimated by referencing geostrophic calculations to surface velocity derived from satellite altimetry, see Methods) shows a consistent pattern at the shelf break and on the inner shelf (Fig. 2e–f), with southward flows (5–20 cm s$^{-1}$) on the eastern flank of the Sabrina Depression and weaker northward flow further west. Our comprehensive observations confirm the presence of a cyclonic gyre on the shelf, consistent with the circulation inferred from limited float data in an earlier study[18], as well as satellite-derived sea surface velocity (Supplementary Fig. 2) and iceberg drift tracks (Supplementary Fig. 4).

Along the continental margin, including the TIS front region, mCDW is limited to the bottom layers in specific deep troughs (Figs. 3, 4; Supplementary Fig. 5a), as first observed by Rintoul et al. [11]. Each trough receives mCDW with different temperatures, with the warmest water (0.24 °C) observed in C-TT (station T5, March 2018, Figs. 4a, 5a) and cooler water (<−0.5 °C) in E-TT (station 43, February 2020, Supplementary Fig. 5a). The temperature of the mCDW arriving at the deep troughs near the ice front is lower (−0.5 to +0.24 °C) than that observed in the depression and at the shelf break (0–0.8 °C) (Figs. 2, 3; Supplementary Fig. 3), suggesting as discussed below that interaction between the thermocline depth and the shallowest sill regulates the

temperature of the water reaching the ice front. Note, however, that the water reaching the ice front is still warm enough to drive rapid basal melt at the deep (>2000 m) grounding line[3]. For example, the warmest water observed in C-TT (Figs. 4a, 5a) would exceed the freezing point by more than 3.7 °C if it can access the grounding line (the in situ freezing point decreases as pressure increases).

Near the continental margin, warm (>−0.5 °C) mCDW is only observed where the water depth exceeds 600–700 m (purple-shaded region in Fig. 3b). In particular, warm mCDW is largely constrained to C-TT and the Sabrina Depression. In contrast, the maximum temperature of mCDW at the bottom of E-TT is cooler (<−0.5 °C), despite depths there being >1000 m (Figs. 1b, 3b; Supplementary Fig. 5a). This suggests that exchange with the depression is more restricted for E-TT than C-TT, consistent with our observations of shallow topography adjacent to E-TT (noting that our survey did not cover all of the regions between E-TT and the depression, Figs. 1b, 3b). The temperature of water reaching the grounding lines is determined not only by the temperature of the water supplied to the trough from the depression but also likely by an interaction between the depth of the thermocline and the height of the shallowest topography (sills and ridges) in the deep troughs and beneath the ice shelf[18–20]. The WW/mCDW interface (i.e., thermocline) is typically at ~500–600 m depth in the Totten region (Figs. 2c, 4; Supplementary Figs. 3, 5a), across which temperature increases with depth by up to ~1.4 °C over ~100 m. At C-TT, where the shallowest measured sills have depths of ~600–700 m, all but the warmest water in the Sabrina Depression can pass through the trough. At E-TT, the shallowest measured sills are shallower (<500–600 m deep), apparently restricting the passage of warm water through the trough. Therefore, the distribution of mCDW along the continental margin suggests that C-TT is the primary route by which warm water reaches the TIS cavity.

## Variability of warm water reaching the TIS cavity
Multi-year hydrographic profiles and mooring time series demonstrate that changes in thermocline depth on a range of timescales are associated with large changes in the thermal forcing available to drive melt of the TIS. Repeat hydrographic stations in C-TT were occupied on four

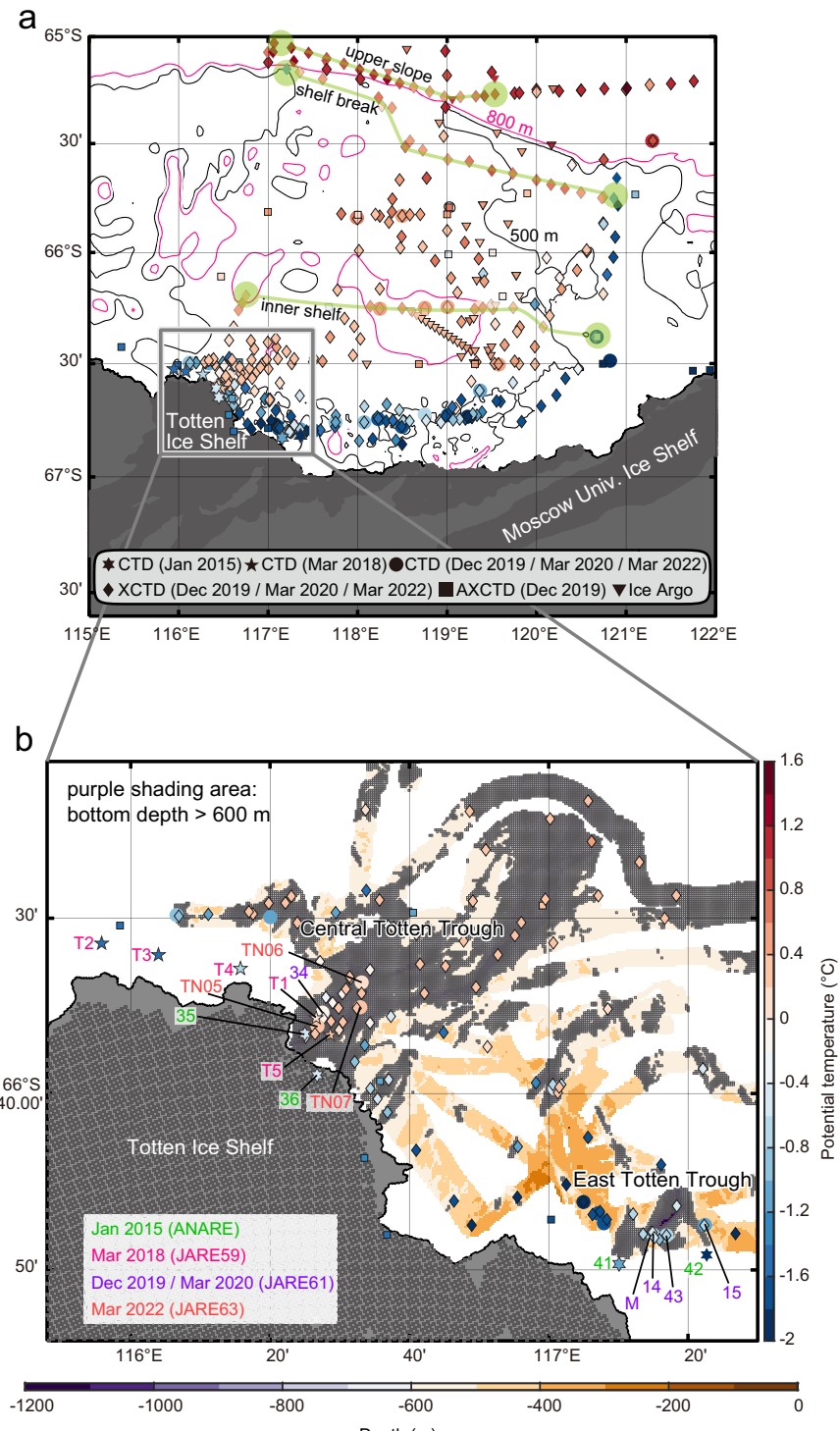

**Fig. 3 | Distribution of warm water in relation to deep topography.** Maps of the potential temperature maximum ($\theta_{max}$, °C) below 200 dbar, corresponding to the potential temperature maximum in the bottom layers if mCDW is present, and the bottom topography. **a** Broad view off the Sabrina Coast and (**b**) close-up view near the Totten Ice Shelf (purple shading area represents the region with bottom depth >600 m derived from the multibeam sonar data). Under-ice positions of the Australian Ice Argo Floats deployed during JARE61 are linearly interpolated. In situ

(not potential) temperature is shown for AXCTD for reference because salinity data were not accurately obtained. In (**a**), the 500-m and 800-m isobaths derived from the gridded bathymetry are shown in black and pink, respectively. In addition, the locations of the upper slope, shelf break, and inner shelf sections shown in Fig. 2 are indicated by the green lines. In (**b**), the CTD stations and mooring site used in Figs. 4, 5 and Supplementary Fig. 5 are indicated with labels ("M" represents the mooring site).

summer cruises (January 2015[11]; March 2018; March 2020; March 2022) (Figs. 3b, 4). The hydrographic profiles show that the maximum temperature of mCDW varies over a wide range (from −0.55 °C to +0.24 °C, Fig. 4a). Salinity profiles also show variable mCDW salinity, and robust

haloclines (hence pycnoclines) at the interface between the surface layer and WW, and between WW and mCDW (Fig. 4b). Overall, vertical profiles with warmer/cooler mCDW are associated with a shallower/ deeper thermocline (Fig. 4a), suggesting that variations in mCDW

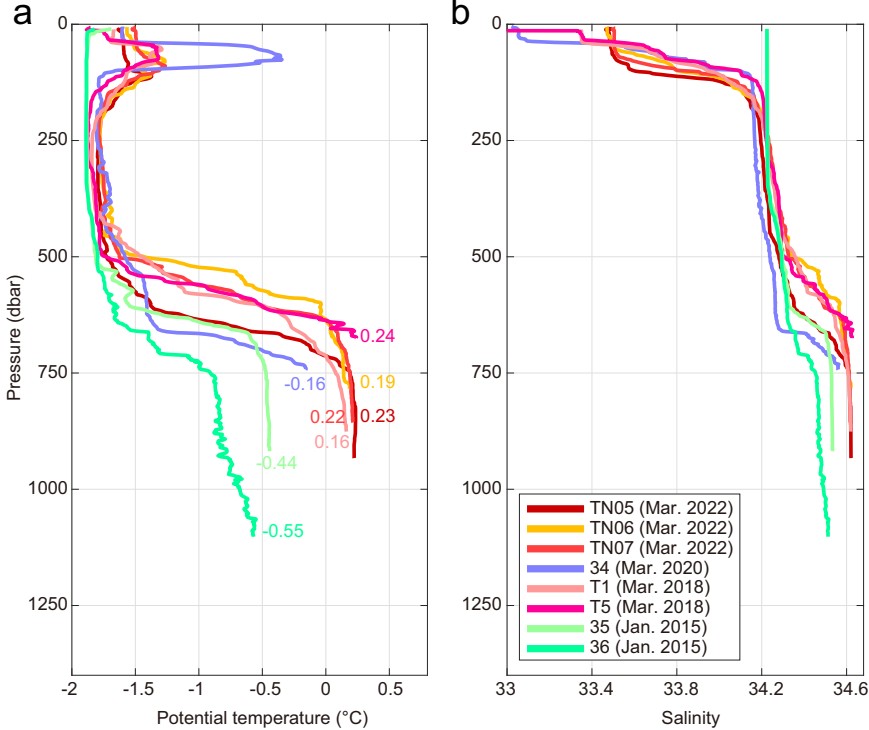

**Fig. 4 | Repeat hydrographic observations in the Central Totten Trough.** Comparisons of repeat CTD (**a**) potential temperature (°C) and (**b**) salinity profiles obtained at the Central Totten Trough (C-TT) in January 2015 (stations 35 and 36, ANARE)[11], March 2018 (stations T1 and T5, JARE59), March 2020 (station 34, JARE61), and March 2022 (stations TN05, TN06, and TN07, JARE63). See the positions of CTD stations in Fig. 3b. In (**a**), numbers beside each profile represent the near-bottom temperature maximum values.

temperature are linked to changes in thermocline depth on an inter-annual timescale. Note that the two coldest observations were obtained in January (2015), while the warmer observations were collected in March (2018, 2020, and 2022), raising the possibility the variations in time reflect seasonal rather than interannual variability. However, the mooring record from E-TT, spanning December 2019 to March 2020 period, shows much smaller variability (±0.1 °C, Supplementary Fig. 5d), and the range between March measurements in different years is also large (−0.16 °C to +0.24 °C).

The time series measurements at E-TT provide further evidence that the interaction between topography and thermocline depth regulates the temperature of water reaching the TIS. Fluctuations in temperature at the mean depth of the WW/mCDW interface are roughly synchronous with those in the near-bottom mCDW layer (Supplementary Fig. 5d), with higher mCDW temperature when the thermocline depth is shallow and vice versa, as seen in Fig. 4a. The unsmoothed moored pressure and temperature records co-vary on tidal frequencies (at semi-diurnal, diurnal, and fortnightly (i.e., spring-neap) cycles, Supplementary Figs. 5–6), indicating that tidal motions drive heaving of the thermocline and fluctuations in mCDW temperature in the trough. Note, however, that we do not find clear evidence of a relationship between zonal wind along the Sabrina Coast and WW/mCDW interface variations (Supplementary Figs. 5–6). This suggests that wind-driven Ekman upwelling/downwelling is not the dominant process driving thermocline depth variations (e.g.[21,22]) in the Totten region, at least at the sub-seasonal scale.

### Evidence of meltwater outflow from TIS cavity

Basal melt caused by inflow of warm mCDW to the ice shelf cavity should be accompanied by outflow of glacial meltwater from the cavity. Previous authors[11,12] speculated that the meltwater outflow might be concentrated on the western side of the ice front, but heavy sea ice prevented sampling of that region. In March of 2018, when the

warmer mCDW was observed at C-TT (0.16–0.24 °C, exceeding the in situ freezing point at -700–800 m depth at the TIS front by more than 2.7 °C, Fig. 4a), we collected hydrographic observations on the previously unsampled western side of the TIS front (stations T1 to T5, see their locations in Fig. 3b), including measurements of the oxygen isotope ratio ($\delta^{18}$O, the ratio of $^{18}$O:$^{16}$O in seawater expressed in the delta notation). Stable oxygen isotopes in seawater are sensitive tracers of freshwater origin in polar waters and are especially sensitive to input of glacial meltwater as it is the primary source of isotopically light freshwater[23–25]. The $\delta^{18}$O along the TIS front was lower throughout the water column in the west than at the central ice front (Fig. 5d), indicating a larger inventory of glacial meltwater in the west. The distribution of glacial meltwater fraction in the WW layer (<500 m depth), calculated by a three-end-member mass balance formulation[25–27], shows an increase from the central ice front (1.1–1.5%, stations T1 and T5) to the western ice front (1.5–1.7%, stations T2 and T3). The WW layer at the western ice front (stations T2 and T3) is also warmer and lower in DO (Fig. 5a, c), and surface waters are fresher (Fig. 5b), than at the central ice front. These signals mainly reflect the mixing of glacial meltwater with warm/low-DO mCDW, as observed, for example, in the Shirase Glacier region[22]. Our direct measurements of glacial meltwater therefore confirm earlier speculation[11,12] that the outflow of meltwater from the TIS cavity is concentrated at the western ice front. Furthermore, the meltwater signal is distributed over a range of depths in the west (Fig. 5), possibly due to different mixing ratios and/or processes, such as boundary mixing and entrainment by buoyant meltwater plume, of glacial meltwater, mCDW, and WW at different depths. Meltwater signals have also been observed over a range of depths at the Pine Island Glacier region[28].

### Simulation of ocean circulation off the Sabrina Coast

Ocean circulation on the Antarctic continental shelf is strongly controlled by the bottom topography. In particular, the pathways of warm

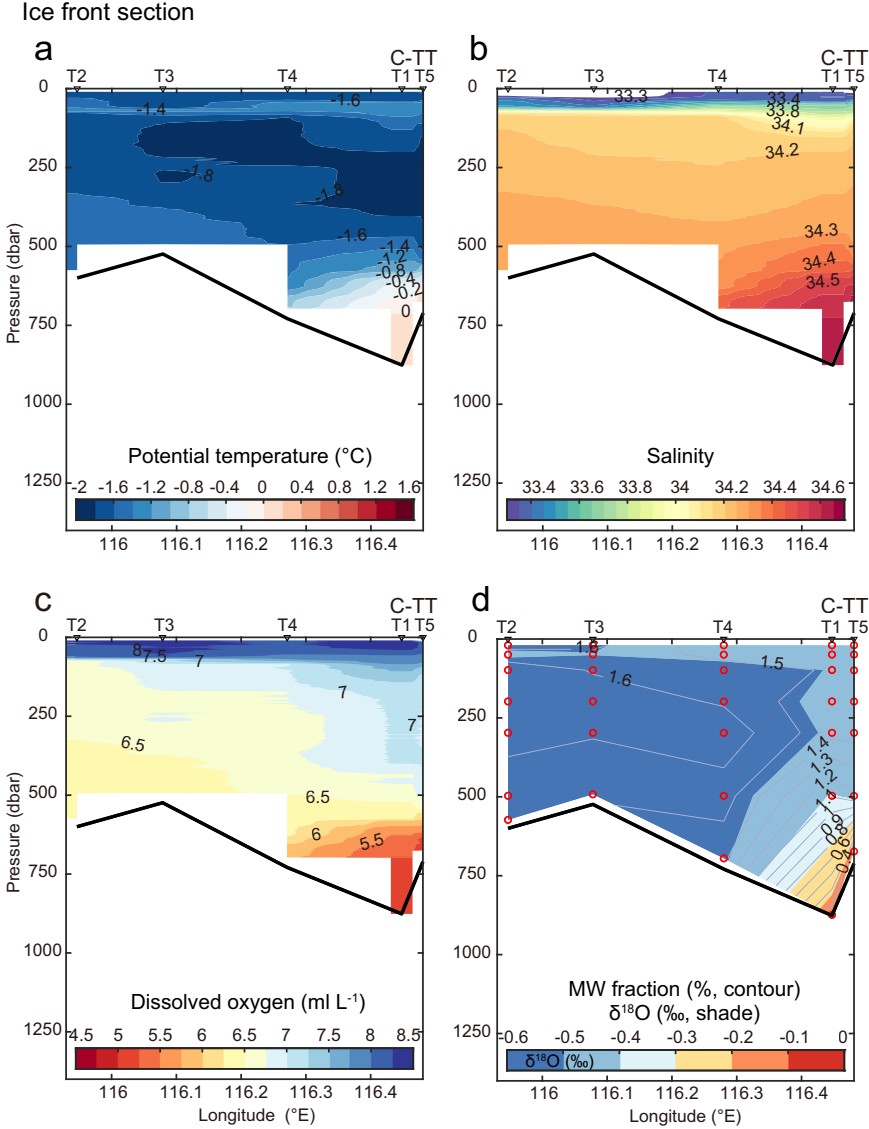

**Fig. 5 | Meltwater signals at the western Totten Ice Shelf front.** Vertical sections of (**a**) potential temperature (°C), (**b**) salinity, and (**c**) dissolved oxygen (ml L⁻¹) and (**d**) stable oxygen isotope ratio ($\delta^{18}$O, ‰, shade) and glacial meltwater fraction (%, contour) at CTD stations (from west to east: T2, T3, T4, T1, and T5, see their positions in Fig. 3b, C-TT: Central Totten Trough) from the western to central Totten Ice Shelf (TIS) front in March 2018 during JARE59. In (**d**), red circles indicate the locations for water samples.

deep water intrusion are sensitive to the representation of the bottom topography. As the bathymetry off the Sabrina Coast had not been well constrained by observations, previous modeling studies may not have represented accurately the pathways of warm water and the resulting ocean–ice shelf interaction in this region. We therefore use our improved bathymetry to configure an ocean-sea ice model (COCO)[29,30] to further investigate the circulation on the continental shelf and to quantify ocean–ice shelf interactions off the Sabrina Coast. The model has a high horizontal resolution of 3–4 km near the Totten Glacier region (see Supplementary Fig. 7 for the model's resolution and Methods for model setup), and thus it can represent the main deep troughs connecting the Sabrina Depression to the TIS cavities (although features in the troughs with spatial scales of a few km are not fully resolved).

The simulated TIS basal melting ($47 \pm 7$ Gt yr⁻¹ for the 1991–2020 climatology, Supplementary Fig. 8) is similar to the satellite-derived estimate ($63.2 \pm 4$ Gt yr⁻¹)[6]. The simulated spatial distributions of near-bottom temperature (Fig. 6a) and circulation (Fig. 6b; see also Supplementary Fig. 9) show that mCDW flows across the shelf break onto the continental shelf mainly at 119–120°E and then circulates in a clockwise direction along the edge of the deep depression (Fig. 6a–c). Note that poleward warm-water inflows would be modulated by the variations in the strength of the westward flowing ASC on the continental slope region[31]. The simulated on-shelf circulation is clockwise, consistent with in situ observations (Fig. 2e,f) and previous model simulations[18,31,32], with a transport of 0.8–1.2 Sv (Fig. 6b, 1 Sv = $1.0 \times 10^6$ m³ s⁻¹). Further offshore, the simulated circulation is characterized by the presence of multiple semi-permanent cyclonic eddies (Supplementary Fig. 9), consistent with the satellite-derived circulation pattern[16] (Supplementary Fig. 2).

In addition, the simulation reproduces the observed distribution of warm mCDW (Fig. 3), with a part of the clockwise circulating mCDW branching toward the TIS via deep troughs at the TIS front, while the rest of the mCDW continues to circulate northward (Fig. 6a, b). The simulation also indicates that while mCDW decreases in temperature and salinity during on-shelf circulation (Fig. 6a, c), when the mCDW accesses the TIS cavity, temperatures remain high enough to induce ice melting, with temperatures exceeding the in situ freezing point at

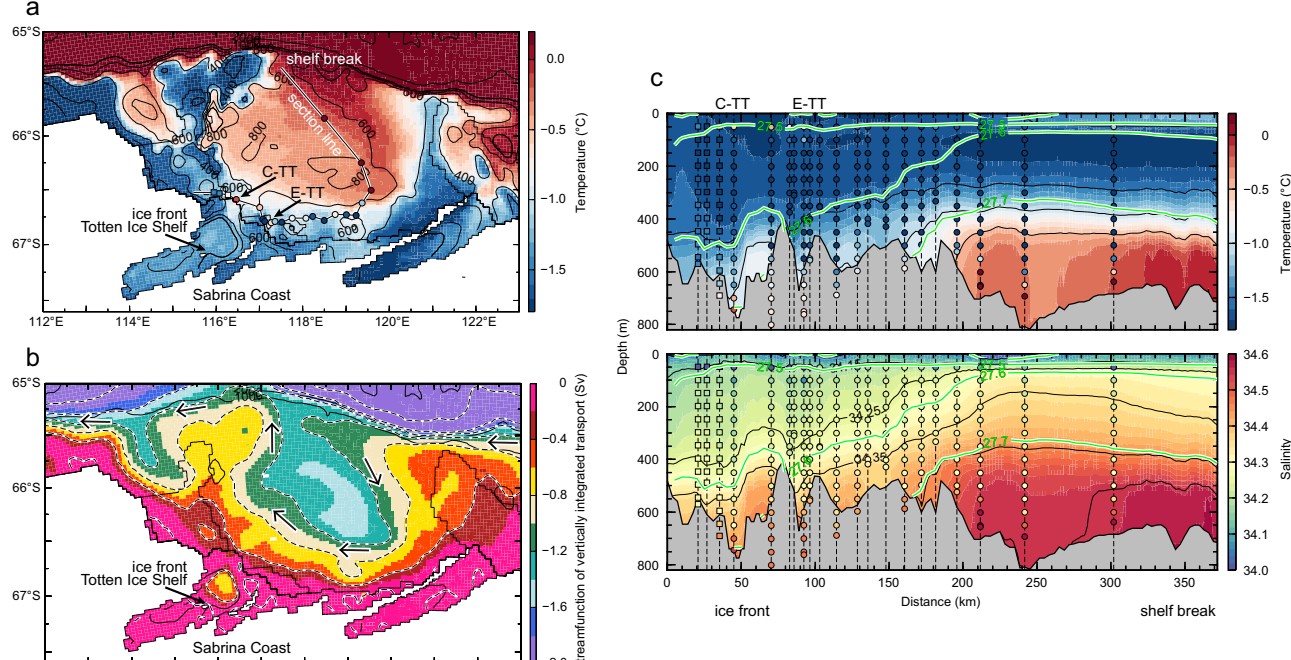

**Fig. 6 | Simulated ocean properties and circulation off the Sabrina Coast.** **a** Ocean temperature on the sea floor (°C). Circles and squares represent CTD observations from JARE61 and JARE59 cruises, respectively. **b** A streamfunction of vertically integrated transport (Sv, 1 Sv = 1.0 × 10⁶ m³ s⁻¹); arrows indicate the flow direction. **c** Vertical sections of ocean temperature (°C) and salinity, from shelf break (right) to the Totten Ice Shelf (TIS) front (left) along the cyclonic on-shelf circulation (the line connecting the observational stations in (**a**)). (C-TT: Central Totten Trough; E-TT: East Totten Trough) The model results are the annual-mean climatology averaged for the period 1991–2020. The observed temperature and salinity values are indicated by the color of the circles at each profile location. Note that, due to the depth gaps between in situ observation and model topography, some CTD data near the TIS front regions are overlaid on the gray-shaded area representing model topography.

~600–800 m depth at the TIS front by more than 1.6–2 °C (Supplementary Fig. 10).

Together with the ocean observations, the model simulation clearly illustrates the pathways of mCDW to and beneath the TIS. Simulated sections along the TIS front (Fig. 7a–c) show that warm/saline mCDW flows into the TIS cavity through the bottom layers of C-TT and E-TT. The most significant mCDW inflow, marked by the thicker mCDW layer along with the bottom-intensified inflow of 4–8 cm s⁻¹, occurs in the bottom layer (deeper than 500–600 m) centered at the eastern side of C-TT. Lower salinity (less dense) waters exit the cavity on the western side of the TIS front. The results from the observations (Figs. 3b, 4, 5) and simulations (Fig. 7a, c) demonstrate that C-TT, directly connecting the TIS cavity and the Sabrina Depression (Figs. 1, 3), is the main conduit by which ocean heat reaches the TIS and drives basal melt.

To test the hypothesis that the deep troughs facilitate access of warm water to the TIS cavity, we performed an additional numerical experiment in which the deep troughs (>500 m depth) in front of the Sabrina Coast were filled in (see Methods for the experimental setup). A comparison of the simulations with and without the deep troughs (hereafter referred to as "CTRL" and "NOTR" cases, respectively) clearly demonstrates the importance of the C-TT as a conduit for ocean heat transport to the TIS cavity. In the CTRL case (Fig. 8a–c), the model shows substantial ocean heat transport of 642 GW into the TIS cavity through C-TT, with a large contribution from inflow of warm/saline mCDW in the bottom layers (Fig. 7a–c). Warm water also reaches the cavities beneath the nearby eastern TIS (eTIS) and western Moscow University Ice Shelf (wMUIS), but the heat transport is relatively small (118 GW at eTIS, 300 GW at wMUIS). In the NOTR case (Fig. 8d–f), little warm reaches the ice front (Supplementary Fig. 11), the inflow of warm deep water is weak, and the ocean heat transport into the three ice-shelf cavities is dramatically reduced throughout the year (annual-

mean ocean heat transports are 111 GW at TIS, 66 GW at eTIS, and 94 GW at wMUIS). These results demonstrate that the deep troughs allow warm water to reach the ice shelf cavities and that C-TT is the main pathway by which warm and high-salinity deep waters from the Sabrina Depression reach the cavity beneath the TIS.

The model simulation furthermore allows us to quantify the vertical circulation beneath the TIS cavity (the so-called ice-pump circulation, Fig. 9) in a model with realistic bathymetry. The simulated transport of mCDW into the TIS cavity of up to ~0.25 Sv (Fig. 9a, b) agrees well with a previously observed transport of ~0.22 Sv[11] based on a snapshot in 2015. During the circulation in the TIS cavity, the mCDW melts the TIS from below, mixes with the resulting glacial meltwater, and becomes lighter (0.1–0.2σθ lower in density). The mixture flows along the base of the ice shelf to the north and finally exits the cavity, reducing the salinity and density of the water column at the western TIS front (Fig. 7b, c). The simulated outflow of glacial meltwater at the western ice front agrees with observations of a larger inventory of glacial meltwater (i.e., lower δ¹⁸O values) in the west (Fig. 5d). In the NOTR case (Fig. 9c), the ice-pump circulation is dramatically weakened since the TIS cavity is occupied by cold water at all depths in the absence of warm water inflows (Supplementary Fig. 11).

## Implications of warm water intrusion toward the TIS
The observations and model simulations highlight multiple factors that favor the transport of warm mCDW to the TIS, as summarized in Fig. 10. Multiple semi-permanent offshore cyclonic eddies (Supplementary Figs. 2, 9) enhance poleward transport of mCDW from the open ocean to the continental slope and shelf break[15,16] (Fig. 2a, b, d, e), increasing the reservoir of heat over the continental slope. The relatively deep shelf break allows warm mCDW to access the continental shelf (Fig. 2b, c, e, f). Warm mCDW fills the bottom layer of the broad Sabrina Depression on the shelf and circulates clockwise around the

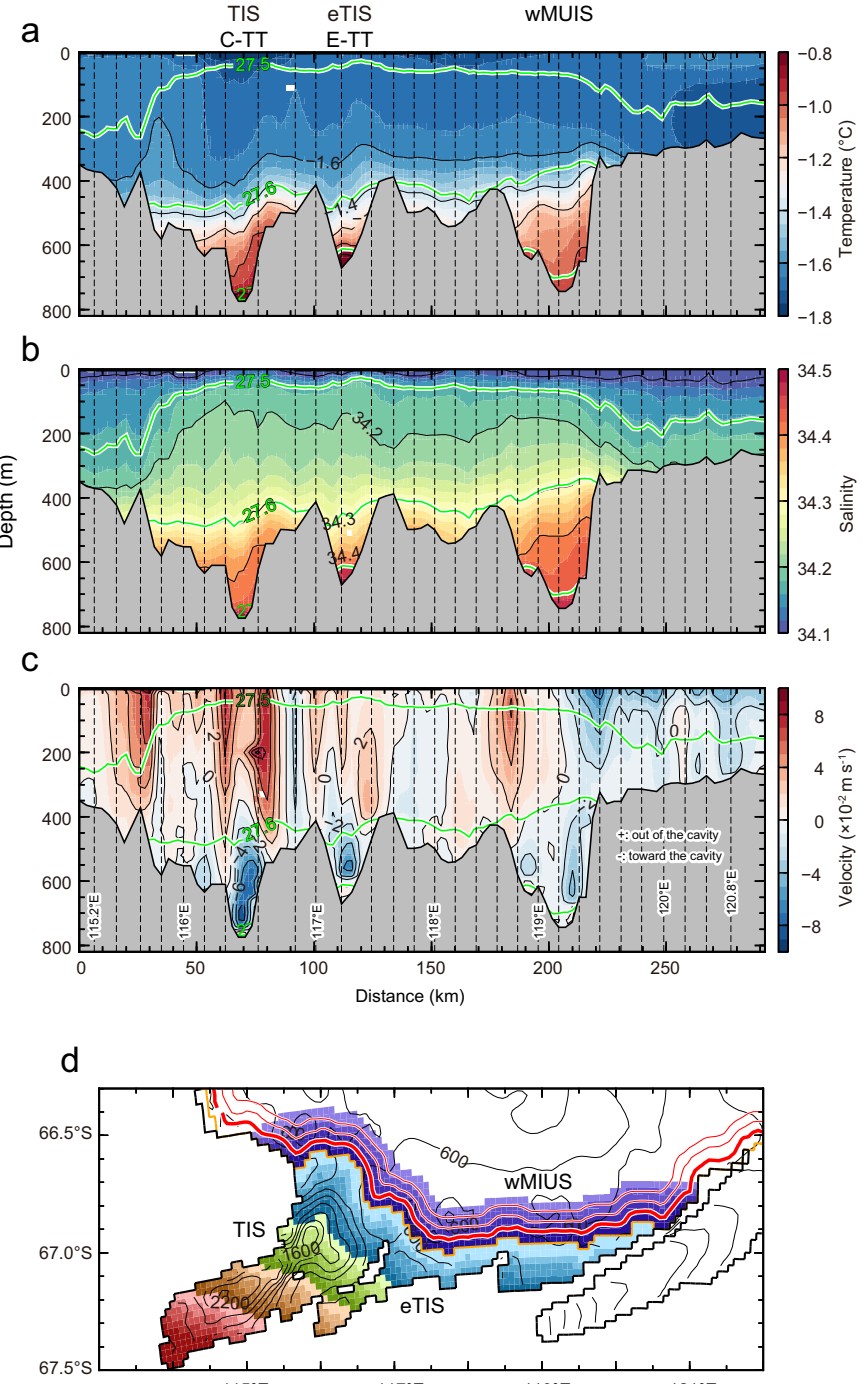

**Fig. 7 | Simulated warm water inflow/meltwater outflow into/out of the Sabrina Coast ice-shelf cavities.** Vertical sections of (**a**) ocean temperature (°C), (**b**) salinity, and (**c**) ocean velocity (m s$^{-1}$) on the section along the Totten Ice Shelf (TIS), eastern TIS (eTIS), and western Moscow University Ice Shelf (wMUIS) fronts (thick red line in (**d**)). The model results are the annual-mean climatology averaged for the period 1991–2020. Green lines in (**a**)–(**c**) show the contours of the potential density anomaly. In (**c**), positive (negative) values in the ocean velocity normal to the section indicate flow out of (toward) the cavity. **d** Coastal and ice-shelf regions were categorized every 5 km from ice-shelf front to calculate overturning circulation in Fig. 9. Red contours show the distance from the ice-shelf front in the open ocean and the thick red line indicates the along-coast section in (**a**)–(**c**). Orange line shows the ice front. In the NOTR case, areas deeper than 500 m in purple regions were set to 500 m.

edge of the depression to the inner continental shelf (Figs. 3a, 6). A branch of this circulating mCDW deflects toward the TIS through narrow and deep troughs that extend to the ice front, with C-TT the primary conduit for ocean heat to reach the ice shelf cavity (Figs. 3b, 6, 7, 8, 9).

The delivery of ocean heat to the TIS cavity depends on not only the temperature but also the thickness of the mCDW layer on the

continental shelf (Figs. 3b, 4a; Supplementary Fig. 5d). In particular, the moored time series and repeat hydrography show that the temperature of mCDW in the deep troughs varies on tidal-to-interannual timescales and suggest that the temperature changes are linked to changes in thermocline depth (Fig. 4a; Supplementary Fig. 5). While the warmest water in the deep part of the Sabrina Depression is blocked from reaching the TIS, sufficiently warm water enters the

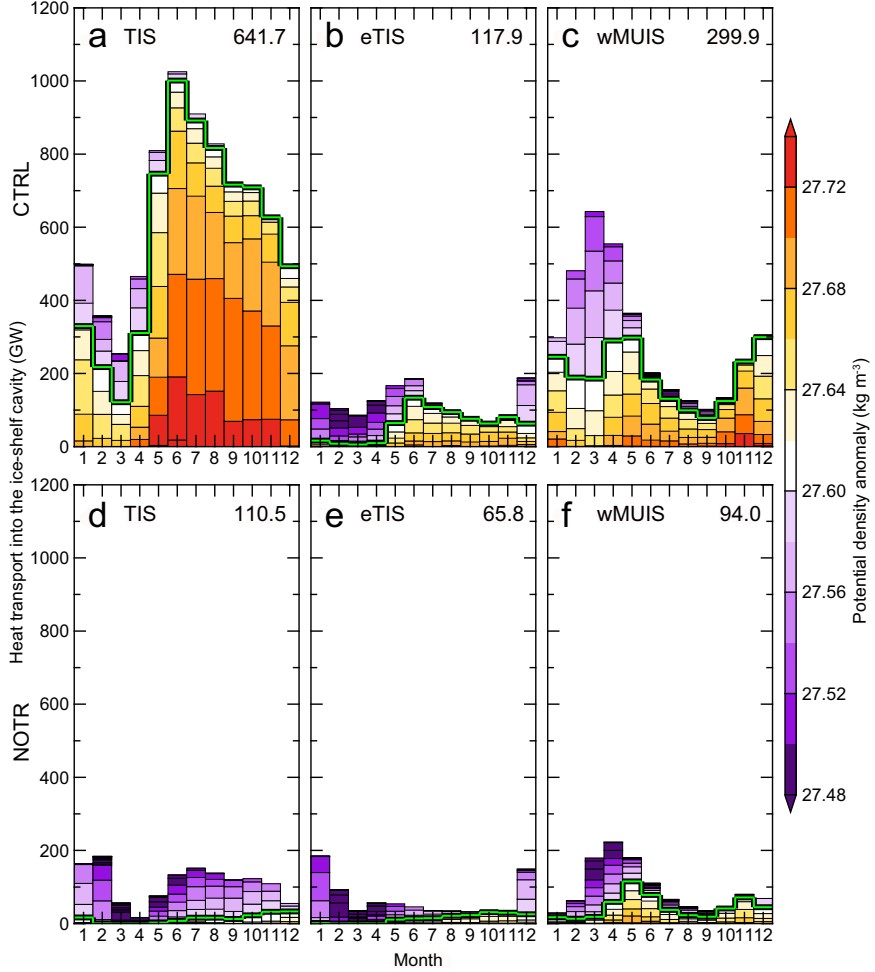

**Fig. 8 | Simulated ocean heat transports into Sabrina Coast ice-shelf cavities.** Seasonal variation in ocean heat transports into the Totten Ice Shelf (TIS), eastern TIS (eTIS), and western Moscow University Ice Shelf (wMUIS) cavities for (**a**–**c**) CTRL and (**d**–**f**) NOTR cases. The reference ocean temperature for ocean heat transport calculation was −2.0°C. The results are the climatology averaged for 1991–2020, calculated from the monthly values of water properties and velocity. The number on the right-upper corner in each panel indicates annual-mean ocean heat transport (GW). Colors show the potential density anomaly and the green line indicates the 27.60 kg m$^{-3}$ isopycnal.

cavity to drive strong basal melt. This, in turn, means that the depth of the thermocline relative to the height of the shallowest sill in the deep trough regulates the delivery of ocean heat to the TIS cavity (Figs. 3b, 4a; Supplementary Fig. 5d). The vertical temperature and salinity profiles on the Totten continental shelf are characterized by a sharp thermocline usually found between ~500 and 600 m depth (Fig. 4; Supplementary Fig. 3). The thermocline on the Totten continental shelf is deeper and sharper than observed on the Amundsen Sea continental shelf in West Antarctica, where the thermocline extends between 300 and 700 m depth[20,33]. Since the depth of the thermocline is similar to the depth of the shallowest sills in the deep Totten Trough, a slight change in the depth of the thermocline results in a relatively large temperature change of mCDW reaching the TIS cavity. Given the observed thermocline strength (up to -1.4 °C per 100 m between 500 and 600 m depth), a vertical displacement of only ~60 m relative to the shallowest sill in the trough could explain the ~0.8 °C range between the years with coldest and warmest mCDW in C-TT (Fig. 4a). Note that this mechanism depends on the relative depths of the thermocline and the shallowest topography along the mCDW pathway and is nonlinear: if the thermocline shoals to be shallower than the blocking topography, warm water will consistently reach the cavity and variability will be damped; the reverse is true if the thermocline deepens to the point where it no longer interacts with the sill.

Various processes may contribute to driving variability in temperature and thermocline depth on the Antarctic continental shelf. Remote or local changes in wind forcing can drive changes in circulation on the slope and shelf, in the exchange between the open ocean and the continental shelf, and in the depth of the thermocline (e.g.[20,21,34–36]), thus altering the temperature and the thickness of the mCDW layer on the shelf. In the Totten region, wind-driven changes in the strength of upwelling over the upper continental slope[37], of the westward flow of the ASC[31], and of the eastward undercurrent on the upper continental slope[18] have been linked to changes in the transport of mCDW onto the Totten continental shelf. Once on the continental shelf, the temperature and thickness of the mCDW layer may be altered by mixing or water mass transformation. For example, changes in buoyancy forcing associated with sea ice production in the Dalton Polynya may also drive changes in stratification and ocean heat supply to the TIS[32,38]. Input of glacial meltwater can increase the temperature and thickness of the warm mCDW layer by inhibiting deep convection that could vent heat from the mCDW layer before it reaches the ice shelf[27]. As observations are at present limited to four summer snapshots, it is not yet possible to determine which of these potential driving factors are most important to driving variability in ocean-driven basal melt of the Totten Glacier.

Glacial meltwater input keeps WW on the Totten continental shelf relatively fresh, inhibiting the formation of Dense Shelf Water (DSW)[27],

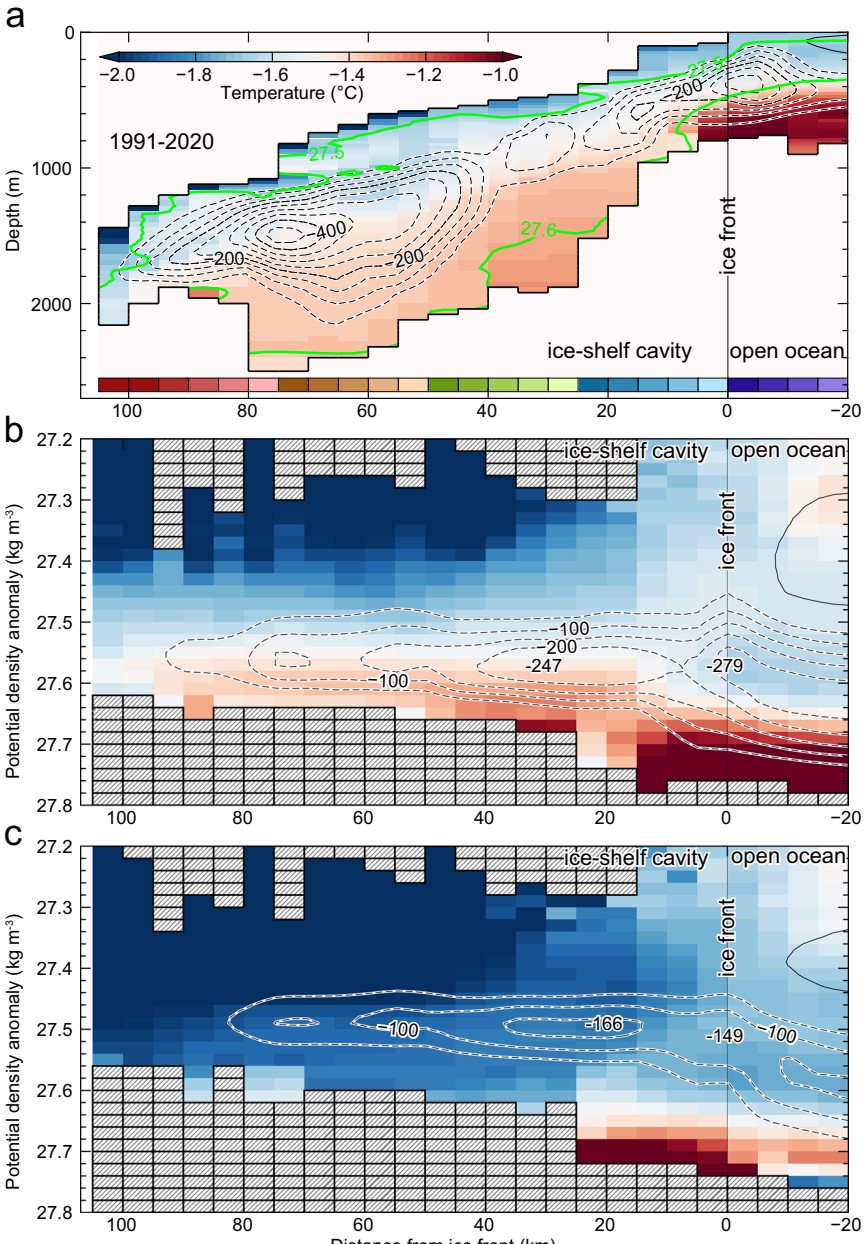

**Fig. 9 | Simulated overturning circulation in the Totten Ice Shelf cavity.** Ocean temperature (°C, color) and overturning circulation (mSv, 1 mSv = 1.0 × 10³ m³ s⁻¹, black solid and dotted contours) in the Totten Ice Shelf (TIS) cavity from the CTRL case in (**a**) distance-depth and (**b**) distance-density domain and (**c**) from the NOTR case in distance-density domain. The horizontal axis is the distance from the ice-shelf front, and positive (negative) values point to the inside (outside) of the cavity (see also Fig. 7d for the area classification). Green lines in (**a**) show the contours of the potential density anomaly. Negative values (dotted lines) of the streamfunctions indicate clockwise circulation. The bottom colorbar in (**a**) corresponds to the areas classified in Fig. 7d. The model results are the annual-mean climatology averaged for 1991–2020, calculated from the monthly values of water properties and velocity.

the precursor of Antarctic Bottom Water. Our observations and simulations show that the meltwater outflow from the TIS cavity is concentrated on the western side of the ice front (Figs. 5, 7a–c). Westward advection of Totten meltwater may affect downstream regions, including Vincennes Bay. The Vincennes Bay Polynya forms a relatively light variety of DSW, which may reflect the impact of both local[39] and remote (e.g. Totten) sources of meltwater.

Satellite observations show that the Totten Glacier has thinned and the grounding line has retreated in recent decades[40,41]. These changes may reflect an increase in ocean heat transport to the TIS cavity, but this hypothesis cannot be tested with observations because oceanographic measurements at the TIS front (ref. 11. and this study)

span a short time interval. However, long-term warming of CDW over the East Antarctic continental slope (warming of 0.8–2.0 °C between 1930–1990 and 2010–2018, between 80 and 160°E)[42] and a significant southward shift in the Southern Boundary of the Antarctic Circumpolar Current, especially off the Sabrina Coast at -120°E[43], may have increased the reservoir of heat available offshore of the Sabrina Coast's ice shelves. If the warmer offshore waters were transported to the deep troughs by the mechanisms described above and illustrated in Fig. 10, these changes in the open ocean may have increased basal melt of the TIS and contributed to the observed thinning and grounding line retreat. In addition, models suggest the Totten is sensitive to increases in oceanic thermal forcing. For example, a recent

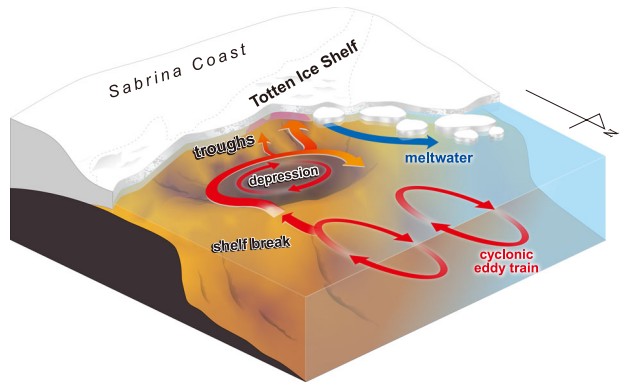

**Fig. 10 | Warm route from offshore to Totten Glacier.** Schematic showing warm water pathways from the open ocean to the Totten Ice Shelf (TIS) cavity. The reservoir of ocean heat over the continental slope is increased where cyclonic eddies carry warm water south on their eastern side. The deep shelf break allows warm modified Circumpolar Deep Water (mCDW) to access the shelf. Warm mCDW pools on the bottom of the Sabrina Depression, which spans most of the Totten embayment. A cyclonic circulation around the rim of the depression carries warm mCDW to the inner shelf, where some of the warm water is diverted through deep troughs to reach the front of the TIS. The warm mCDW enters the ice shelf cavity by bottom-intensified flows in the deep troughs, especially in Central Totten Trough (C-TT), where the largest ocean heat transport occurs (shown by the broad arrow toward the cavity). Glacial meltwater outflow is concentrated on the western side of the ice front.

model study using projected changes in ocean temperature, salinity, and velocity using CMIP6 emission scenarios as forcing, found that warm water (−0.5–1 °C) accesses the sub-ice shelf cavity along the eastern ice front, driving sustained retreat of the eastern grounding zone[44].

Our study provides valuable insights into the physical processes that control the melt rate of the Totten Glacier, now and in the future. In particular, our observations from the TIS front suggest that changes in depth of the thermocline drive variability in thermal forcing on a broad range of timescales (tidal to interannual). This further implies that changes in future forcing or regional circulation that alter the thickness and warmth of the mCDW layer will drive changes in basal melt. While our study provides comprehensive observations spanning nearly the full Totten embayment and robust evidence of temporal variability in the inflow of warm mCDW to the TIS cavity, further observations are needed to better understand the drivers of mCDW variability and the exposure of the Totten Glacier to ocean-driven melt, now and in the future. The large interannual variability observed in the temperature of mCDW reaching the TIS (Fig. 4a) highlights the need for long-term monitoring that resolves seasonal-to-interannual variability of the oceanic thermal forcing driving basal melt of the TIS. In particular, time series of the temperature, thickness, and transport of warm water along the pathways connecting the open ocean to the ice shelf cavity are needed to determine the sensitivity of the Totten Glacier with its potentially large contribution to sea level rise, to future changes in natural and anthropogenic forcing.

## Methods
### Comprehensive shipborne and airborne observations around the TIS
In December 2019 and February/March 2020, we made comprehensive ship- and airborne observations to obtain hydrographic and bathymetric data from the upper continental slope/shelf break, continental shelf, and ice front regions around the TIS on the Sabrina Coast from the icebreaker "*Shirase*" (AGB-5003). This observational campaign was conducted during the 61st Japanese Antarctic Research Expedition (JARE61), under the "Research of Ocean-ice BOundary

InTeraction and Change around Antarctica" (ROBOTICA) project of the National Institute of Polar Research. In March 2018 (JARE59) and March 2022 (JARE63), we made the shipborne hydrographic and bathymetric observations at the TIS front and continental shelf regions under the same project.

For hydrographic observations, we measured surface-to-near-bottom profiles of temperature, conductivity, pressure, and DO using a conductivity–temperature–depth profiler (CTD, SBE19, Sea-Bird Scientific, USA) with a DO sensor (SBE43, Sea-Bird Scientific, USA) (stars and circles in Fig. 3). Additionally, expendable CTD probes (XCTD, Tsurumi-Seiki Co., Ltd., Japan) were deployed (diamonds in Fig. 3). In December of 2019 only, airborne expendable CTD probes (AXCTD, Tsurumi-Seiki Co., Ltd., Japan) were deployed from the *Shirase*'s helicopter (squares in Fig. 3).

### Bathymetric data off Totten Glacier
We acquired multibeam bathymetric data in uncharted areas off Totten Glacier with a 20 kHz SeaBeam3020 system (L3 Communications ELAC Nautik) installed on the icebreaker *Shirase*. The operation was conducted as a part of the JARE61 and JARE63 Science Program. The multibeam data were acquired during December 2019, February/March 2020, and March 2022 and were provided by Japan Coast Guard (JCG) for scientific use (Supplementary Fig. 1a). The sounder has 205 beams with a transducer of 2 (transmission) by 2 (receiving) degrees. The seawater sound velocity correction was conducted using real-time data from the surface water velocity meter and the vertical profiles collected by CTD and XCTD observations. The HIPS and SIPS software (CARIS, Ltd., Fredericton, Canada) was used for data processing to remove extreme depth variations, mainly from the outer edges of the multibeam swaths.

The multibeam sonar data from JARE observations were compiled with two sets of previously collected multibeam data (Supplementary Fig. 1a); the EM120 (Kongsberg) installed on the icebreaker *N.B. Palmer* (survey number 14-02)[13,45,46] and the EM122/EM710 (Kongsberg) installed on the R/V *Investigator* cruise (survey number IN2017-V01)[47]. The *N.B. Palmer* data was obtained through Marine Geoscience Data System (MGDS) hosted by Lamont-Doherty Earth Observatory of Columbia University, then manually processed with the same methods as applied to the *Shirase* data. The *Investigator* data was provided in a processed format by the CSIRO National Collections and Marine Infrastructure.

To obtain the digital terrain model for physical oceanographic modeling in this study, we further utilized ship-based single-beam echo sounding data obtained during JARE57, JARE58, and JARE59 cruises (Bathy 2010, SyQwest Inc.), and 2014/15-V2 cruise (AA2014) of RSV *Aurora Australis* (SIMRAD EK60, Kongsberg) (Supplementary Fig. 1b). Data archived in the NOAA NCEI database with the following survey numbers were also used in this study; JARE29L4, TH83, OB3B, AU9407, AU9604, AU9807, OB1, OB3B, NBP0101, ELT46, and ELT50 (Supplementary Fig. 1b). In addition, we adopted point estimated depth data derived from AXCTD and AXBT observations (Supplementary Fig. 1b). Data gaps were filled using the global ocean and land terrain model GEBCO2020[48], in which estimated seafloor topography derived from satellite altimetry mainly covers the study area. The 1-km resolution bathymetric map was finally generated using all the above data (Supplementary Fig. 1c). Data were processed using the GMT system[49], and bathymetric data were gridded using a weighted nearest-neighbor algorithm and surface algorithm in which adjustable tension continuous curvature splines were used.

### Calculation of absolute ocean velocity from hydrographic and satellite-derived dynamic ocean topography (DOT) data
The DOT is the sea surface height (SSH) relative to the geoid height and consists of the baroclinic (steric) and the barotropic (bottom pressure term) components. We employ a monthly mean DOT dataset (with

0.2° × 0.2° grid spacing) over the Southern Ocean (spatially seamless even in sea-ice-covered regions, but except for regions close to the Antarctic Continent or landfast sea ice attached to the coast), derived from multiple satellite radar altimeter datasets[16] (data period for this study is from January 2011 to December 2020, Supplementary Fig. 2), to examine the ocean circulation from offshore to the continental shelf regions. With this DOT dataset, the ocean surface geostrophic velocity can be calculated even in the sea-ice-covered region (vectors in Supplementary Fig. 2). Note that tides do not alias the DOT-derived velocity calculation because we apply a FES2004 tidal correction[50] to the SSH before obtaining the DOT.

The DOT-derived velocity, which can be obtained at the sea surface only, reflects both vertically varying baroclinic and vertically homogeneous barotropic components. Meanwhile, hydrographic XCTD measurements provide the geostrophic velocity vertical profile, containing only the baroclinic component and not the barotropic component. We then obtain the vertical profiles of absolute velocity, including both baroclinic and barotropic components, by shifting the whole vertical profile of geostrophic velocity estimated with XCTD hydrographic profiles to such that the geostrophic velocity at the surface agrees with the DOT-derived velocity. We used the monthly mean DOT-derived velocity in February 2020, the closest to the observation period, for calculating the absolute velocity shown in Fig. 2. We determined the DOT-derived velocity closest to where the geostrophic velocity was calculated with the search range of 0.1°.

## Estimation of glacial meltwater fraction

Salinity and oxygen isotope ratio $\delta^{18}O$ can be used as conservative parameters to quantify the respective freshwater contribution from glacial and sea-ice meltwaters. Assuming that observed salinity and $\delta^{18}O$ ($S_{obs}$ and $\delta^{18}O_{obs}$) can be explained as a mixture of three end-members of modified Circumpolar Deep Water (mCDW), meteoric water (i.e., glacial meltwater), and sea-ice meltwater, the three-component mass balance is described by:

$$f_{sim} + f_{met} + f_{mCDW} = 1 \tag{1}$$

$$f_{sim} S_{sim} + f_{met} S_{met} + f_{mCDW} S_{mCDW} = S_{obs} \tag{2}$$

$$f_{sim}\delta^{18}O_{sim} + f_{met}\delta^{18}O_{met} + f_{mCDW}\delta^{18}O_{mCDW} = \delta^{18}O_{obs} \tag{3}$$

where $f_{sim}$, $f_{met}$, and $f_{mCDW}$ are the fractions of endmembers of sea-ice meltwater, meteoric water, and mCDW, respectively[25–27]. In this study, we used the following endmember values: sea-ice meltwater ($S_{sim} = 6.2$, $\delta^{18}O_{sim} = 1.2$ ‰, ref. 27), meteoric water ($S_{met} = 0$, $\delta^{18}O_{met} = -30$ ‰, ref. 27), and mCDW ($S_{mCDW} = 34.69$, $\delta^{18}O_{mCDW} = -0.06$ ‰, derived from our observations on the upper slope). Standard deviation of our $\delta^{18}O$ measurement was estimated to be 0.016‰. The largest uncertainty comes from the $\delta^{18}O$ endmember of meteoric meltwater. The melt line drawn on $\delta^{18}O$-salinity diagram for the data in Fig. 5d points toward a $\delta^{18}O$ of $-32$ ‰ at zero salinity from the values of mCDW, which is consistent with the inland surface accumulation value[27]. Assuming the uncertainty of 5% (ref. 25), the error in meteoric water fraction would be about 0.5% for a typical seawater of the upper layer. Although this error range is seemingly large, practical error in spatial difference in meltwater fraction should be much less among nearby stations because of the systematic nature of the error assumption.

## Numerical model and model configuration

We utilized an ocean–sea ice model, COCO (https://ccsr.aori.u-tokyo.ac.jp/~hasumi/COCO/), with an ice-shelf component[29,36]. The horizontal grid of the model was discretized on an orthogonal, curvilinear,

horizontal coordinate system, and we can regionally enhance the horizontal resolution around the Totten Glacier region by placing the two singular points on the longitude of the focal area (75°S, 120°E and 0°, 120°E). With this treatment, the horizontal resolution near the Totten Glacier region was 3–4 km, keeping the model domain to the circumpolar Southern Ocean with an artificial solid boundary north of 43.2°S (depending on the longitude).

The vertical coordinate system of the ocean model was a $z$-coordinate, with a vertical grid spacing of 5 m (4 grid levels) for the surface layers, 20 m (59 grid levels) for the 20–1200-m range, 40 m (30 grid levels) for the 1200–2400-m range, and 100 m (31 grid levels) for the 2400–5500-m range. The maximum ocean depth in the model was set to 5500 m. The regionally high horizontal resolution enabled us to capture the realistic coastline and icescape. We used the General Bathymetric Chart of the Oceans[51] as the background bottom topography for the Southern Ocean. We replaced the bottom topography around the Totten Glacier region in the following steps:

(1) The bottom topography in the region (112–126°E, 64–68°S) was replaced with the detailed topography derived in this study and described in the Methods section for bathymetry above (Note that multibeam sonar bathymetric data from the 2021/22 season was not incorporated into the model bathymetry due to the unavailability at the time when we prepared the model configuration.);

(2) The topography under the ice shelves and the ice-shelf draft were calculated from BedMachine Antarctica[5];

(3) In the coastal region within 20 km of the ice-shelf front, we blended our compiled data and BedMachine's topography, selecting the deeper topography of the two.

The icescape in this region was characterized by well-developed landfast sea ice[52,53]. Because sea ice processes are strongly controlled by the positions of the icescape, we introduced the landfast sea ice areas into the model with a constant thickness (5 m) for ice-shelf grid cells, as in our previous studies[36]. We assumed a steady-shaped ice shelf in the horizontal and vertical directions in the model. A three-equation scheme with velocity-independent coefficients[54] for the thermal and salinity exchange velocities ($\gamma_t = 1.0 \times 10^{-4}$ m s$^{-1}$ and $\gamma_s = 5.05 \times 10^{-7}$ m s$^{-1}$) was used to estimate the freshwater flux at the base of the ice shelves.

Near the artificial northern boundary or north of 40°S, water properties (temperature and salinity) were restored to the monthly climatology of the World Ocean Database 2018[55,56] throughout the water column, with a damping time scale of 10 days. Outside of the focal region where the horizontal resolution is larger than 4.3 km (see Supplementary Fig. 7), we restored the sea surface salinity to the monthly mean to avoid unrealistic model drift in coarse-resolution regions. The model was driven by daily surface boundary conditions consisting of surface winds, air temperature, specific humidity, downward shortwave and longwave radiation, and freshwater flux. Daily atmospheric boundary conditions were calculated from the ERA5 dataset[57], and the bulk formulas for wind stress and sensible and latent heat fluxes were used in the model[58]. We spun up the model from a motionless field with the observed properties for 33 years using the 1950s' conditions (three cycles with 1951–1960 forcing and then 3 years with 1951 forcing) and then performed a hindcast simulation for the period from 1951–2021 with interannually variable forcing ("CTRL" case).

An additional experiment of the NOTR case was performed to quantify the roles of deep troughs in front of the ice-shelf cavities. In the NOTR case, the areas deeper than 500 m along the Sabrina Coast (purple areas in Fig. 7d) were set to 500 m (artificially filling the deep troughs). We performed a simulation of NOTR case for the period 1990–2020. The 1990's initial condition for the NOTR case was derived from the CTRL case. In both CTRL and NOTR cases, the present study used climatology averaged over 1991–2020.

## Data availability

In situ shipborne and airborne observation data used in this study is available at https://data.aad.gov.au/metadata/au1402 (CTD-ANARE, the 2014/15 season), https://ads.nipr.ac.jp/dataset/A20230206-001 (CTD-JARE59, the 2017/18 season), https://ads.nipr.ac.jp/dataset/A20230206-002 ($\delta^{18}$O water sample-JARE59, the 2017/18 season), https://ads.nipr.ac.jp/dataset/A20230206-003 (CTD-JARE61, the 2019/20 season), https://ads.nipr.ac.jp/dataset/A20210708-003 (XCTD-JARE61, the 2019/20 season), https://ads.nipr.ac.jp/dataset/A20220616-001 (AXCTD-JARE61, the 2019/20 season), https://ads.nipr.ac.jp/dataset/A20230206-004 (Mooring timeseries-JARE61, the 2019/20 season), https://ads.nipr.ac.jp/dataset/A20230131-005 (CTD-JARE63, the 2021/22 season), https://ads.nipr.ac.jp/dataset/A20230131-006 (XCTD-JARE63, the 2021/22 season). The bathymetric dataset compiled and used in this study is available at https://ads.nipr.ac.jp/dataset/A20230206-005. The satellite-derived dynamic ocean topography data[16] is available at http://www2.kaiyodai.ac.jp/~mizobata/datashare.html. The numerical model simulation results will be available from K.K. upon requests.

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

## Acknowledgements

The authors are deeply grateful to the officers, crews, and scientists on board icebreaker *Shirase* for their help with field observations. We also acknowledge technical support related to airborne observations provided by D. Duncan, G. Ng, and D. Buhl. The model simulations were carried out using DA system and Earth Simulator in JAMSTEC. This work was supported by Grants-in-Aids for Scientific Research (JP20H04961, JP20K12132, JP17H06316, JP17H06317, JP17H06322, JP17H06323, JP17H01615, JP21H04918, JP17H04710, JP21H04931, JP19K12301, JP20H04979, JP17H01157, JP20H05707, JP21H03587, JP22H01337, JP22H05003, JP20H04970, JP21H01201, and JP21K13989) of the Ministry of Education, Culture, Sports, Science and Technology, the Science Program of Japanese Antarctic Research Expedition (JARE) as Prioritized Research Project, National Institute of Polar Research (NIPR) through Project Research KP-303 and KP-306, the Center for the Promotion of Integrated Sciences of SOKENDAI, and the Joint Research Program of the Institute of Low Temperature Science, Hokkaido University (19S007, 20S008, 21S007, and 22S012). K.K. was supported by MEXT-Program for the advanced studies of climate change projection (SENTAN) Grant Number JPMXD0722681344. M.F. was supported by "Challenging Exploratory Research Projects for the Future" grant from Research Organization of Information and Systems (ROIS). Y.N. was supported by Inoue Science Research Award from Inoue Science Foundation. S.R. and E.V.W. were supported by grant funding from the Australian Government as part of the Antarctic Science Collaboration Initiative program and by the Centre for Southern Hemisphere Oceans Research. J.S.G. was supported by NSF project OPP-2114454 and NASA's Cryosphere Program under grant 80NSSC22K0387; D.D.B. and J.S.G. were supported by NSF project OPP-1543452 and the G. Unger Vetlesen Foundation. We finally acknowledge the use of imagery provided by services from NASA's Global Imagery Browse Services (GIBS), part of NASA's Earth Observing System Data and Information System (EOSDIS).

## Author contributions

D.H. conceived this study. D.H. and T.T. designed the field observations. D.H. led and conducted shipborne observations during JARE59. S.A. and T.T. led the field observation campaign during JARE61. K.Y., Y.Nakayama, K.O., T.I., and Y.A. conducted the JARE61 field observations. K.I.O. contributed to mooring data acquisition. J.S.G. conceived of the airborne observations for JARE61, designed the initial airborne survey, and provided receiver electronics; D.D.B. supported the airborne observations. S.R. and E.V.W. contributed the Argo floats and data. D.S. produced pre-bathymetric grid data for observation planning. M.F. compiled and processed the available bathymetric data. Y.Nogi and K.S. contributed to the bathymetry data acquisition. K.K. performed the numerical simulation. K.M. analyzed the satellite observation data. S.A. led the Science Program of the Japanese Antarctic Research Expedition (JARE) as Prioritized Research Project (ROBOTICA project) and measured the sampled water. D.H., S.A., K.K., and S.R. wrote the paper, and all authors discussed the results and commented on the manuscript.

## Competing interests

The authors declare no competing interests.
