## [Peer Review File · Nature Communications]

On-Shelf Circulation of Warm Water Toward the Totten Ice Shelf in East AntarcticaEditorial Note: The figure on page 21 in this Peer Review File has been amended to remove third-party material where no permission to publish could be obtained.

REVIEWER COMMENTS

Reviewer #1 (Remarks to the Author):

This is a very good, well-written paper presenting interesting results. It concerns the Totten Ice Shelf in East Antarctica, a less famous (but nonetheless important) sister of the Amundsen Sea ice shelves. The processes revealed are of interdisciplinary interest, being relevant to oceanographers and glaciologists wanting to understand what leads to ice shelf mass loss, and/or the biogeochemical or climate impacts of the resulting meltwater. The primary results are the insights about the circulation revealed by a new swath bathymetry data set, and by hydrographic surveys, supported by a cutting-edge high resolution model.

The paper describes and synthesises some great observations, but would benefit from more clearly identifying what is new/surprising, and why it matters, as it goes along. For example the section on the pathways of warm water was clear and well described, but I was left wondering whether any of the features were different to previous papers. It's fine if there's nothing new there, but in that case make it clearer that the section is just setting the scene for subsequent sections?

I think the key results are those stated in lines 220 to 225. These are clear, important and valuable. If the authors confirm my understanding, probably these key results should be stated more strongly at the end of the paper as the take-home message?

Recommendations for strengthening:

1. Many of the figures have multiple (fairly unrelated) panels and this means that the fonts on many of the panels are impossible to read. There are 16 figure panels, which in a longer-format journal would probably be presented in 10-12 separate figures; this makes it feel like a longer paper trying to squeeze into a high profile format. I recommend cutting down the number of figures to those that are essential to support the key message(s) of the paper, and enlarging them so that fonts are legible (e.g. contour labels, axis labels) and the figures themselves are useful to readers.
2. The new bathymetry data are a major new result of your work, I think? This might be sold more strongly. Identify clearly what is new in the section on the updated bathymetry. So much literature is cited here that it is difficult to decipher what the new bathymetry has added to our knowledge.

Minor suggestions:

Line 30 and 32 – are the glacial troughs in the first sentence the same as the deep passages in the second sentence? If so, use the same terminology throughout.

Line 42 – I wasn't sure what it means by a basin being vulnerable to oceanic thermal forcing? Is this an ocean basin? Or part of the ice shelf or ice sheet?

Line 43. In what sense is the "expansive ice behind" the glacier? Or is it behind the ice shelf?

Line 47. Is there another source of oceanic thermal forcing (to justify calling this the strongest)?

Line 80. Repeats what was said in lines 66-67

Line 95. I wasn't sure what was meant by "during at least the last ice retreat timing"?

Line 96. Seem not seems

Line 94-94. Why does this the bathymetric data support this? I couldn't follow the reasoning.

Line 101. How do you know that these are eddies and not just the section cutting across a meandering along-slope current?

Line 169. In where?

Line 172. Did you consider using the O18 measurements to calculate the meltwater distribution?

Line 186. This would benefit from rephrasing: "characterized with an order of 5-10 cm s⁻¹ of ocean currents"

Line 206. I couldn't follow "circulates over the Dalton Polynya from autumn to winter". Why over? Can you circulate from one season to another?

Line 208-211. I couldn't follow this sentence. How does an inflow suggest increasing mCDW?

Line 219. More CDW than what?

Is the westward advection of meltwater a surprise? Don't we expect it from previous (observational and model) studies?

Lines 240-246. These model studies are interesting but not really related to the discussion of your own results, so I cannot see why they are mentioned at all. The need for long-term mooring data is based on these published studies, not your own work. It means that the final paragraph of the paper is a bit speculative and not really based on the strong results of your own paper. I think it would be better to end the paper with a clear statement of the take-home messages based upon your own work (observations and model).

Figure 1. It would make it easier for readers if both panels had the same colorbar. Is this colorbar colorblind-accessible?

Figure 2. Panels a-c are too small to do justice to the observations. Since one of the strengths of your paper is your combined / updated bathymetry, it would be good to add this beneath the CTD sections?

Shouldn't conservative temperature be used rather than potential temperature, following TEOS10?

Figure 3. Panels and fonts are too small to read.

Figure 4. Panel d is a super schematic but too small – I can't read the text. Maybe it's just my old eyes!

Maps 1a, 1b, 2d, 3a, 4a are all different regions/projections/aspect ratios. It would be helpful to aim for more consistency to help your readers.

Karen J. Heywood

Reviewer #2 (Remarks to the Author):

Review of: On-shelf circulation of warm water toward the Totten Ice Shelf, East Antarctica
Authors: Daisuke Hirano, et al.

This manuscript documents new observations collected over the Antarctic continental shelf in the Sabrina Coast region in front of the Totten Ice Shelf. Specifically, a multi beam sonar survey provides details of deep glacial troughs that control the circulation around and under the Totten Ice Shelf. These bathymetric observations are coupled with a mooring time series and high-resolution numerical simulations to update the lateral circulation and heat transport pathways over the continental shelf.

First, I thank the authors for a well written manuscript. This is important field work and the new observations will undeniably improve the ability to model heat and tracer transports in this region of the Antarctic continental shelf as well as improve predictions of future change. However, my

opinion is that the results in this manuscript do not rise to the level of a Nature Communications publication. In particular, many of the key points have already been described in previous high-impact publications by some of the co-authors, including both observational (Rintoul et al. 2016, Hirano et al. 2021) and numerical (Nakayama et al. 202) studies. More importantly, the figures in this short manuscript do not reflect a deep analysis of the data acquired during the recent field work. In the absence of evidence of warm water under and around Totten, the results in this manuscript would be truly novel. However, based on recent work, I was hoping to see a more thorough discussion of the physical processes that impact heat transport over the continental shelf. My feeling is that a longer-format article in a disciplinary journal would provide a better opportunity for the authors to fully explore the exciting data set that they have acquired. Additional details that support this opinion are provided below.

Major comments

1) Figure 1 shows the major advance in this study, which is the identification of deep troughs in front of Totten. While the multi-beam data is impressive, to a certain extent, the presence of this deep trough was already apparent from the hydrographic section in Rintoul et al. (2016) — their station 36 was > 1000-m showing large heat content. Additionally, the multi-beam data is unable to capture the full meridional extent of these troughs so there remains some ambiguity about how the inflow is connected to the Sabrina Depression and ultimately the shelf break.

2) Were there any current measurements made during the field program, either with an ADCP or via current meters on the mooring? The abstract of this manuscript discusses the importance of heat transport and circulation pathways, but there are no velocity fields provided from the observations, and only the barotropic streamfunction from the numerical model is provided in the main text of the manuscript. It would be nice to see the analysis go beyond a simple comparison of hydrographic sections between the model output and the numerical model. In particular, there is a nice time series of T/S and CDW thickness variability — are the same temporal scales captured by the numerical model?

3) As a related point, the abstract states, “These results provide insight into the pathways and processes regulating heat transport to the TIS, ...” First, I felt that the authors could have been more explicit about how the new bathymetry modifies either the transport pathways or the heat flux towards Totten, as compared to previous modeling efforts. There are no quantitative estimates of heat fluxes or transports in this manuscript. The mooring time series is another example where a deeper analysis would have been appreciated. Rather than simply plotting the time series, the paper would be strengthened by a consideration of what processes set this variability. Perhaps looking at a frequency spectra would help attribute changes to short-term surface wind and buoyancy forcing (appreciating that the forcing might be poorly constrained) or eddies. The study by Webber et al. (Nat. Comm. 2017) is a nice example of this approach applied to mooring data in front of Pine Island Glacier.

4) Supplementary Figure 8: This is probably the most important figure in the study. It would have been nice to see this in the main text. However, there are still some confusing aspects in this figure. First, the caption only says “cumulative transports” — is this a heat or a volume transport or both? The figure has units of (mSV) labeled, but it is hard to judge a magnitude because there are no numbers on the y-axis. It would be nice to see a similar panel with heat transports. The caption says the transports are binned by density, but it is hard to judge the vertical structure with this presentation. The traditional way to produce an overturning circulation like this is to simply plot the volume transport in density space — it would be nice to see this figure. It would also make it easier to see if/how the overturning transport closes across the different sides of this box.

Minor comments

Line 69: “We also examine the variability in the temperature of mCDW reaching the TIS, which is directly related to the magnitude of TIS melting ...” A reference linking the observed melt rates at Totten to the on-shelf temperature content would be helpful. I expect this to be true based on detailed studies in West Antarctica, but are there sufficient observations to confirm this on the

Sabrina Coast?

Line 94: "While the Sabrina Depression has very likely not been affected by glacial erosion during at least the last ice retreat . . ." Please add a citation here or at least justify these comments.

Pathways of warm water section: Even if ADCP data is not available from the cruise, it would be helpful to see section of the geostrophic velocities. While the barotropic component may be uncertain, either a clearly stated assumption about a level or no motion or use of the numerical model output would provide a clearer picture of the circulation. Related to this is that there is no discussion of the overturning circulation on the shelf in this manuscript and how that might be influenced by the new bathymetry observations.

Line 127: "This suggests that TTR1 is the main conduit by which ocean heat is delivered for TIS melting." Without showing the velocity fields, I am not sure how the authors reach this conclusion – this can not be deduced by only looking at the temperatures. Both the circulation and heat content contribute to the total heat flux.

Spatiotemporal variability section: There is no discussion or speculation about possible causes for the sub-seasonal variability. Tides are mentioned at one point, but the authors do not provide information about the expected magnitude of the tides in this region.

Line 179: "showing a reasonable model representation of ice shelf-ocean interaction in this region." Is the parameterization for ice-shelf melt in this model tuned to agree with this value?

Simulation section: Overall, the modeling work is very nice and provides a useful complement to the observations. However, it feels like the analysis here is somewhat superficial because of the length of the manuscript. The only real comparison in the main text are two mean sections of T and S shown in Figure 4b. There are some significant differences between the thickness of the CDW layer in these panels, but this is not discussed in detail. Can these be attributed to seasonal fluctuations, uncertainty in surface forcing in the model, other?

Figure 4c: Units are not provided for the streamfunction.

Response to the specific comments and suggestions from Reviewer #1

Dear Prof. Karen Heywood,

We very much appreciate your positive assessment and constructive/helpful comments. Your comments were invaluable for improving our manuscript. We have addressed all your comments/suggestions and have substantially modified the paper with the enhanced in-situ observations and simulations (Koji Saito, Japan Coast Guard, was newly added as a co-author due to the addition of the latest observation data.) Our response to each comment is written in red italic and we have highlighted the changes made to the revised manuscript. We now believe that the revised manuscript meets the criteria for publication in Nature Communications.

REVIEWER COMMENTS

Reviewer #1 (Remarks to the Author):

This is a very good, well-written paper presenting interesting results. It concerns the Totten Ice Shelf in East Antarctica, a less famous (but nonetheless important) sister of the Amundsen Sea ice shelves. The processes revealed are of interdisciplinary interest, being relevant to oceanographers and glaciologists wanting to understand what leads to ice shelf mass loss, and/or the biogeochemical or climate impacts of the resulting meltwater. The primary results are the insights about the circulation revealed by a new swath bathymetry data set, and by hydrographic surveys, supported by a cutting-edge high resolution model.

The paper describes and synthesises some great observations, but would benefit from more clearly identifying what is new/surprising, and why it matters, as it goes along. For example the section on the pathways of warm water was clear and well described, but I was left wondering whether any of the features were different to previous papers. It's fine if there's nothing new there, but in that case make it clearer that the section is just setting the scene for subsequent sections?

- *Previous papers relevant to this study have already provided some of the following observational findings: poleward (eddy-induced) warm water inflow across the shelf break (Nitsche et al., 2017; Hirano et al., 2021) and subsequent warm water access/distribution to the Sabrina Depression (Silvano et al., 2019) and Totten Ice Shelf (TIS) front (Rintoul et al., 2016). However, the linkages between the key process mentioned above have not been defined due to the insufficient (spotty) in-situ hydrographic and bathymetric data in the region. Thus, to date, there has been no integrated description/knowledge of warm-water distribution and circulation from offshore to the TIS front regions in association with realistic bathymetry, one of the*

essential pieces of information for understanding ocean circulation. Therefore, we conducted comprehensive shipborne and airborne observations from the upper slope, continental shelf, to the TIS front, aiming for an integrated understanding of the warm water distribution/circulation that would control the magnitude of oceanic thermal forcing for the TIS basal melt.

The above is the novelty associated with the first submitted version. Furthermore, in the revised manuscript we have included newly acquired in-situ observation data (bathymetric and hydrographic data) from the 2021/22 season (JARE63) in the Totten region (see figures below), which we believe further strengthens the novelty of this paper on warm water distribution and circulation towards TIS cavity controlled by deep bathymetric features. Our multi-beam sonar survey, for the first time, revealed the relationship with detailed bathymetry. Thus, we could clarify the comprehensive warm water circulation from the upper slope to the TIS front regions. Of particular note is that the TIS cavity is connected to the continental shelf (depression) by the TTR1's deeper topography of at least >600–700 m, and warm water flows along this deep channel toward the TIS. This indicates, together with the simulation results, that TTR1 is the main conduit by which ocean heat is delivered for TIS melting.

In addition, by building a high-resolution model incorporating the realistic updated bathymetry data, we could simulate water mass distribution and ocean circulation, consistent with in-situ observations, even near the TIS front. The model simulation further allowed us to illustrate the picture of vertical circulation beneath the TIS cavity (i.e., ice-pump circulation) for the first time. (Line: 63-71; 200-243)

We believe the above points are the major novelty of this study and advance the field from previous studies.

I think the key results are those stated in lines 220 to 225. These are clear, important and valuable. If the authors confirm my understanding, probably these key results should be stated more strongly at the end of the paper as the take-home message?

- *Thank you for your valuable suggestion. We have modified the take-home message to one established on our new findings (circulation and temporal variability of the mCDW inflowing into TIS) and have highlighted the necessity for long-term monitoring of the ocean state in the Totten region. (Line: 276-289)*

Recommendations for strengthening:

1. Many of the figures have multiple (fairly unrelated) panels and this means that the fonts on many of the panels are impossible to read. There are 16 figure panels, which in a longer-format journal would probably be presented in 10-12 separate figures; this makes it feel like a longer paper trying to squeeze into a high profile format. I recommend cutting down the number of figures to those that are essential to support the key message(s) of the paper, and enlarging them so that fonts are legible (e.g. contour labels, axis labels) and the figures themselves are useful to readers.

- *Based on the comment, we have reorganized the overall figure structure by decomposing the multiple unrelated panels, moving figures from the Supplementary Information, and creating new figures. The revised manuscript now consists of ten figures. We have also enlarged the figures for ease of reading (Figures 1-10)*

2. The new bathymetry data are a major new result of your work, I think? This might be sold more strongly. Identify clearly what is new in the section on the updated bathymetry. So much literature is cited here that it is difficult to decipher what the new bathymetry has added to our knowledge.

- *We agree with your comment. We believe that new bathymetry data obtained by multibeam sonar is one of the major highlights of this paper (major new result). To*

further strengthen this, we have added the in-situ bathymetric data newly acquired in the 2021/22 season (JARE63) in this revised manuscript. With the first bathymetry dataset from the multibeam sonar around the TIS, we could reveal almost the full extent of TTR1, as the main conduit by which ocean heat is delivered for TIS melting and have significantly improved our understanding of the warm water circulation toward the TIS, which is controlled by the bathymetry. (Line: 74-95; 131-142; 226-234)

Minor suggestions:

Line 30 and 32 – are the glacial troughs in the first sentence the same as the deep passages in the second sentence? If so, use the same terminology throughout.

- *No, they are not the same. We have deleted the word “passage” to avoid misinterpretation throughout the revised paper.*

Line 42 – I wasn’t sure what it means by a basin being vulnerable to oceanic thermal forcing? Is this an ocean basin? Or part of the ice shelf or ice sheet?

- *It is not an “ocean” basin but a “subglacial” basin below the ice base of the ice sheet/shelf. For clarity, we have modified this sentence. (Line: 40-42)*

Line 43. In what sense is the “expansive ice behind” the glacier? Or is it behind the ice shelf?

- *Based on your comment, we have modified this part by deleting “expansive ice behind.” (Line: 40-42)*

Line 47. Is there another source of oceanic thermal forcing (to justify calling this the strongest)?

- *It is seasonal, but Antarctic Surface Water is also a strong oceanic thermal forcing for the ice-shelf melting (e.g., Aoki et al., 2022). Based on your comment, we have modified this sentence to avoid misunderstanding. (Line: 45-46)*

Line 80. Repeats what was said in lines 66-67

- *Thank you for pointing this out. We have deleted the redundant section (Line 58-60).*

Line 95. I wasn’t sure what was meant by “during at least the last ice retreat timing”?

Line 96. Seem not seems

Line 94-94. Why does this the bathymetric data support this? I couldn’t follow the reasoning.

- We cannot show the timing relationship between the glacial activity and topographic formation using data or proper references. We have retained minimal wording in this part because it does not inherently affect the conclusions of the study. (Line: 94-95)

Line 101. How do you know that these are eddies and not just the section cutting across a meandering along-slope current?

- Thank you for pointing this out. For this revision, we created the absolute ocean velocity (Fig. 2d-f) from the geostrophic velocity calculation, referenced to the satellite-derived surface ocean velocity, following the same method as Hirano et al. (2021). Since the observations along the upper slope section (Figs. 2a, d) do not follow the isobath, the results might partly include features such as a meandering slope current; however, at least the warm water core at ~119.5E corresponds to the downstream of the southward flowing region of the eastern half of the West Sabrina Eddy. The subsequent southward inflow is still observed in the shelf break and inner shelf sections east of 119E (Fig. 2b, c, e, f), where warm water widely spreads near the seafloor. In other words, it is an important observational fact that at least the offshore eddies contribute to the poleward warm-water transport across the shelf break, followed by the continuous southward warm water inflow to the Sabrina Depression in the inner shelf. (Line: 97-114)

Line 169. In where?

- We have deleted this sentence in the revised manuscript because we described the ice-pump circulation along with the simulation results.

Line 172. Did you consider using the O18 measurements to calculate the meltwater distribution?

- Thank you for the comment. We have added a brief description of the glacial meltwater distribution estimated from a three-end-member mass balance formulation (Meredith et al., 2008; Silvano et al., 2018). (Line: 190-193)

Line 186. This would benefit from rephrasing: “characterized with an order of 5-10 cm s⁻¹ of ocean currents”

- Thank you for pointing this out. We have modified this sentence. (Line: 217-220)

Line 206. I couldn't follow “circulates over the Dalton Polynya from autumn to winter”. Why over? Can you circulate from one season to another?

Line 208-211. I couldn't follow this sentence. How does an inflow suggest increasing mCDW?

- *For the simulation results, we decided to remove the discussion on seasonal variability and focus on the mean fields (climatology) in this revision because the discussion on seasonal variations could not be compared and verified with observations and included some speculation. Therefore, we have deleted these parts.*

Furthermore, as already mentioned above, we have enhanced simulation results along the ice-front section (Fig.8) and in the TIS cavity (Fig.9), which is the additional novelty of the paper. (Line: 226-243)

Line 219. More CDW than what?

- *Thank you for pointing this out. We have modified this sentence. (Line: 247-249)*

Is the westward advection of meltwater a surprise? Don't we expect it from previous (observational and model) studies?

- *The westward advection itself is not unexpected, as Antarctic Coastal Currents are generally westward, but the meltwater signal concentrated to the west originating from the TIS cavity was only speculated based on hydrographic data (Rintoul et al., 2016; Silvano et al., 2017).*

Additionally, this study offers a multifaceted evaluation from oxygen isotope ratio data and simulation results, which indicate that the meltwater exits the cavity along the TIS base northward, making the western TIS front region fresh/less dense. This is a better presentation of meltwater outflow from TIS and an advance from speculations by previous studies. (Line: 188-193; 226-243)

Lines 240-246. These model studies are interesting but not really related to the discussion of your own results, so I cannot see why they are mentioned at all. The need for long-term mooring data is based on these published studies, not your own work. It means that the final paragraph of the paper is a bit speculative and not really based on the strong results of your own paper. I think it would be better to end the paper with a clear statement of the take-home messages based upon your own work (observations and model).

- *We agree with your comment. As described above, we have modified the take-home message section to the one based on our new findings (circulation and temporal*

variability of the mCDW inflowing into TIS) and have highlighted the necessity for long-term monitoring of the ocean state in the Totten region. (Line: 276-289)

Figure 1. It would make it easier for readers if both panels had the same colorbar. Is this colorbar colorblind-accessible?

- *Based on your comment, we have made Figure 1 easier to read. Additionally, the colorbar used for Figure 1 is commonly used in Marine Geology (“Haxby” colormap of GMT software). Furthermore, we checked whether the main topographic features, such as deep Totten troughs, are discernable with a color-blind simulator (<https://pilestone.com/pages/color-blindness-simulator-1>), and we confirmed they are discernable with this color space.*

Figure 2. Panels a-c are too small to do justice to the observations. Since one of the strengths of your paper is your combined / updated bathymetry, it would be good to add this beneath the CTD sections?

- *Revised Figure 2 differs from the previous one because we have reorganized the overall figure structure. Current Figure 2 consists of temperature and absolute ocean velocity sections. We also added ocean bottom topography extracted from the multibeam sonar dataset for each panel.*

Shouldn't conservative temperature be used rather than potential temperature, following TEOS10?

- *We used the potential temperature in this study, considering the consistency with the ocean-sea ice-ice shelf model (Kusahara and Hasumi, 2013).*

Figure 3. Panels and fonts are too small to read.

- *Revised Figure 3 differs from the previous one, but we have made all figures easier to read.*

Figure 4. Panel d is a super schematic but too small – I can't read the text. Maybe it's just my old eyes!

- *We are happy with your compliment (and sorry for the inconvenience). The schematic picture is now Figure 10 (with no multiple panels). We have also made the text larger so that readers would be able to read it easily.*

Maps 1a, 1b, 2d, 3a, 4a are all different regions/projections/aspect ratios. It would be helpful to aim for more consistency to help your readers.

- Thank you for your comment. We have tried to unify the regions/projections/aspect ratios of maps shown in Figures 1, 3, and 7 as much as possible.

Karen J. Heywood

Response to the specific comments and suggestions from Reviewer #2

We appreciate your constructive/helpful comments very much. Your comments were invaluable for improving our manuscript. We have considered almost all your comments/suggestions and have substantially modified the paper with the enhanced in-situ observations and simulations (Koji Saito, Japan Coast Guard, was newly added as a co-author due to the addition of the latest observation data.) Our response to each comment is written in red italic and we have highlighted the changes made to the revised manuscript. We now believe that the revised manuscript meets the criteria for publication in Nature Communications.

Reviewer #2 (Remarks to the Author):

Review of: On-shelf circulation of warm water toward the Totten Ice Shelf, East Antarctica

Authors: Daisuke Hirano, et al.

This manuscript documents new observations collected over the Antarctic continental shelf in the Sabrina Coast region in front of the Totten Ice Shelf. Specifically, a multi beam sonar survey provides details of deep glacial troughs that control the circulation around and under the Totten Ice Shelf. These bathymetric observations are coupled with a mooring time series and high-resolution numerical simulations to update the lateral circulation and heat transport pathways over the continental shelf.

First, I thank the authors for a well written manuscript. This is important field work and the new observations will undeniably improve the ability to model heat and tracer transports in this region of the Antarctic continental shelf as well as improve predictions of future change. However, my opinion is that the results in this manuscript do not rise to the level of a Nature Communications publication. In particular, many of the key points have already been described in previous high-impact publications by some of the co-authors, including both observational (Rintoul et al. 2016, Hirano et al. 2021) and numerical (Nakayama et al. 2021) studies.

- *Previous papers relevant to this study have already provided some of the following observational findings: poleward (eddy-induced) warm water inflow across the shelf break (Nitsche et al., 2017; Hirano et al., 2021) and subsequent warm water access/distribution to the Sabrina Depression (Silvano et al., 2019) and Totten Ice Shelf (TIS) front (Rintoul et al., 2016). However, the linkages between the key process mentioned above have not been defined due to the insufficient (spotty) in-situ hydrographic and bathymetric data in the region. Thus, to date, there has been no integrated description/knowledge of warm-water distribution and circulation from offshore to the TIS front regions in association with realistic bathymetry, one of the*

essential pieces of information for understanding ocean circulation. Therefore, we conducted comprehensive shipborne and airborne observations from the upper slope, continental shelf, to the TIS front, aiming for an integrated understanding of the warm water distribution/circulation that would control the magnitude of oceanic thermal forcing for the TIS basal melt.

The above is the novelty associated with the first submitted version. Furthermore, in the revised manuscript we have included newly acquired in-situ observation data (bathymetric and hydrographic data) from the 2021/22 season (JARE63) in the Totten region (see figures below), which we believe further strengthens the novelty of this paper on warm water distribution and circulation towards TIS cavity controlled by deep bathymetric features. Our multi-beam sonar survey, for the first time, revealed the relationship with detailed bathymetry. Thus, we could clarify the comprehensive warm water circulation from the upper slope to the TIS front regions. Of particular note is that the TIS cavity is connected to the continental shelf (depression) by the TTR1's deeper topography of at least >600–700 m, and warm water flows along this deep channel toward the TIS. This indicates, together with the simulation results, that TTR1 is the main conduit by which ocean heat is delivered for TIS melting.

In addition, by building a high-resolution model incorporating the realistic updated bathymetry data, we could simulate water mass distribution and ocean circulation, consistent with in-situ observations, even near the TIS front. The model simulation further allowed us to illustrate the picture of vertical circulation beneath the TIS cavity (i.e., ice-pump circulation) for the first time. (Line: 63-71; 200-243)

We believe the above points are the major novelty of this study and advance the field from previous studies.

More importantly, the figures in this short manuscript do not reflect a deep analysis of the data acquired during the recent field work. In the absence of evidence of warm water under and around Totten, the results in this manuscript would be truly novel. However, based on recent work, I was hoping to see a more thorough discussion of the physical processes that impact heat transport over the continental shelf. My feeling is that a longer-format article in a disciplinary journal would provide a better opportunity for the authors to fully explore the exciting data set that they have acquired. Additional details that support this opinion are provided below.

- *The novelty and advance of this paper are noted above. Furthermore, we have performed the following additional analyses for this revision: the estimation of the absolute ocean velocity from the geostrophic and satellite-derived surface velocity (Fig. 2d-f) and spectrum (Fig. S4) and trend (Fig. 6c) investigations of the mooring time series. Please take further note of the response below for details.*

Major comments

1) Figure 1 shows the major advance in this study, which is the identification of deep troughs in front of Totten. While the multi-beam data is impressive, to a certain extent, the presence of this deep trough was already apparent from the hydrographic section in Rintoul et al. (2016) — their station 36 was > 1000-m showing large heat content. Additionally, the multi-beam data is unable to capture the full meridional extent of these troughs so there remains some ambiguity about how the inflow is connected to the Sabrina Depression and ultimately the shelf break.

- *Indeed, the presence of a deep trough with a depth of >1000 m at the TIS front and warm water access to the trough had already been observed (Rintoul et al., 2016). However, the study was based on just-below bathymetry data from a single beam sonar; thus, the details of the spatial structure of the Totten trough were unknown. In contrast, our first*

multibeam-surveyed bathymetric data near the TIS reveals the detailed spatial structure of the Totten trough, especially the most significant TTR1. However, as you pointed out, there may still be ambiguity regarding the topographic connection between the Totten trough and the Sabrina Depression/shelf break.

This revised manuscript includes newly acquired in-situ observation data (bathymetric and hydrographic data) from the 2021/22 season (JARE63) in the Totten region, which we believe have overcome the concern about ambiguity. We have revealed almost the full extent of TTR1 as the main conduit by which ocean heat is delivered for TIS melting and have significantly improved our understanding of the warm water circulation toward the TIS, which is controlled by the bathymetry. Specifically, we successfully showed that the TIS cavity directly connects with the Sabrina Depression (and to the shelf break) by the TTR1's deeper topography of at least >600–700 m. (Line: 74-95; 131-142; 226-234)

2) Were there any current measurements made during the field program, either with an ADCP or via current meters on the mooring? The abstract of this manuscript discusses the importance of heat transport and circulation pathways, but there are no velocity fields provided from the observations, and only the barotropic streamfunction from the numerical model is provided in the main text of the manuscript. It would be nice to see the analysis go beyond a simple comparison of hydrographic sections between the model output and the numerical model. In particular, there is a nice time series of T/S and CDW thickness variability — are the same temporal scales captured by the numerical model?

➤ *There are no available ocean current data either from the mooring or ADCP. Based on the comments, we have carried out the current best practice to estimate the absolute ocean velocity on the upper slope, shelf break, and the inner shelf of the Sabrina Depression sections (Fig. 2d-f) from the geostrophic velocity calculation, referenced to the satellite-derived surface ocean velocity, following the same method as Hirano et al. (2021). The results show a cyclonic circulation, with warm water flowing southward in the eastern half of the Sabrina Depression and northward in the western half (Fig. 3), reinforcing the observational evidence for the on-shelf cyclonic circulation of the warm water. (Line: 97-114)*

Furthermore, the observation of the warm-water cyclonic circulation is consistent with the simulation results (Fig. 7).

With the high-resolution model incorporating the realistic updated bathymetry data, we could simulate the water mass distribution and ocean circulation, consistent with in-situ

observations, even near the TIS front. We believe the enhanced observations of the bathymetry and hydrography near the TIS allow us to compare the results from in-situ observations and numerical simulations for the first time, which in itself is a simple but significant advance. (In other words, until now, it has not been possible to compare even “simple” observation-simulation results.) (Line: 63-71; 200-243)

In addition, we decided to remove the discussion on seasonal variability and focus on the mean fields (climatology) in the revised manuscript. Instead, a separate numerical simulation paper (in preparation) will present an in-depth discussion of variability, including a timescale with more than a seasonal scale (Kusahara et al., in prep.).

3) As a related point, the abstract states, “These results provide insight into the pathways and processes regulating heat transport to the TIS, ...” First, I felt that the authors could have been more explicit about how the new bathymetry modifies either the transport pathways or the heat flux towards Totten, as compared to previous modeling efforts. There are no quantitative estimates of heat fluxes or transports in this manuscript. The mooring time series is another example where a deeper analysis would have been appreciated. Rather than simply plotting the time series, the paper would be strengthened by a consideration of what processes set this variability. Perhaps looking at a frequency spectra would help attribute changes to short-term surface wind and buoyancy forcing (appreciating that the forcing might be poorly constrained) or eddies. The study by Webber et al. (Nat. Comm. 2017) is a nice example of this approach applied to mooring data in front of Pine Island Glacier.

➤ We have added in-situ observations from the 2021/22 season in the Totten region to strengthen our original novelty. Our high-resolution model, incorporating realistic updated bathymetry data, could simulate water mass distribution and ocean circulation, consistent with in-situ observations, even near the TIS front. The model simulation allowed us to illustrate a picture of vertical circulation beneath the TIS cavity for the first time. Integrating results from the realistic simulation and in-situ observations, we found the TTR1 as the main conduit by which ocean heat is delivered for TIS melting and significantly improved our understanding of the warm water circulation toward the TIS, which is controlled by the successive deep channels from the upper slope, depression, to the TIS cavity. This is the main progress from previous studies.

Based on your comments, we additionally performed the spectral analysis of the temperature time series and ERA5 wind data (Fig. S4) to examine the relationship

between inflowing warm water property into the TIS and tides or wind forcing. If we define trends over the mooring period (~3 months) as ranges of seasonal variability, it is reasonable to conclude that the range of mCDW temperature reflects interannual variability rather than seasonal one. In addition, we conclude, from the frequency analysis, a contribution from the wind-driven Ekman upwelling/downwelling on the WW/mCDW interface (i.e., mCDW thickness) variation (e.g., Ohshima et al., 1996; Hirano et al., 2020) is not evident in the Totten region, at least below the seasonal timescale. Certainly, Webber et al. (2017) is an excellent example, but their conclusions are established on investigations with long-term five-year mooring records. In contrast, our time series data are shorter than the seasonal window (summertime only). Furthermore, wind/buoyancy forcing typically has seasonality; therefore, it is difficult to examine the relationship between warm-water properties and wind/buoyancy forcing in a sub-seasonal timescale. (Line: 150-185)

When we obtain more than year-round time series data, discussing variability with more than a seasonal scale will be optimal through comparison and integration with model results.

4) Supplementary Figure 8: This is probably the most important figure in the study. It would have been nice to see this in the main text. However, there are still some confusing aspects in this figure. First, the caption only says “cumulative transports” — is this a heat or a volume transport or both? The figure has units of (mSV) labeled, but it is hard to judge a magnitude because there are no numbers on the y-axis. It would be nice to see a similar panel with heat transports. The caption says the transports are binned by density, but it is hard to judge the vertical structure with this presentation. The traditional way to produce an overturning circulation like this is to simply plot the volume transport in density space — it would be nice to see this figure. It would also make it easier to see if/how the overturning transport closes across the different sides of this box.

➤ *Thank you very much for your constructive comments. In this revision, we removed results and discussion of seasonal variability to focus more on the mean fields (climatology). To take account of your suggestion on the overturning circulation, we have added new figures for the overturning circulations under/near the ice-shelf cavity (Fig. 9) and along TIS front temperature/salinity/ocean velocity sections (Fig. 8). These new figures allow us to clearly show ice-pump circulation and the connection to ocean circulation/condition over the continental shelf region. (Line: 226-243)*

In addition, we considered the seasonal-to-interannual variability in ice-ocean interaction and ocean circulation in the model to be addressed. Thus, we are working on a study that will provide more details about this (Kusahara et al. in prep).

Minor comments

Line 69: “We also examine the variability in the temperature of mCDW reaching the TIS, which is directly related to the magnitude of TIS melting ...” A reference linking the observed melt rates at Totten to the on-shelf temperature content would be helpful. I expect this to be true based on detailed studies in West Antarctica, but are there sufficient observations to confirm this on the Sabrina Coast?

- *Thank you for the comment. Rignot et al. (2002) showed a linear relationship between the basal melt rates vs oceanic thermal forcing (T minus in-situ freezing point). This might address your comment on the circumpolar view, but it might not be valid for the Totten region because of insufficient in-situ observations, at least when Rignot et al. (2002) was published. Thus, we have modified this sentence to avoid misunderstanding. (Line: 61-63)*

[redacted]

Rignot et al. (2002, Science)

Line 94: “While the Sabrina Depression has very likely not been affected by glacial erosion during at least the last ice retreat . . .” Please add a citation here or at least justify these comments.

- *We cannot show/suggest the timing relationship between glacial activity and topographic formation using data or proper references. We have retained minimal wording in this part because it does not inherently affect the conclusions of the study. (Line: 94-95)*

Pathways of warm water section: Even if ADCP data is not available from the cruise, it would be helpful to see section of the geostrophic velocities. While the barotropic component may be uncertain, either a clearly stated assumption about a level or no motion or use of the numerical model output would provide a clearer picture of the circulation. Related to this is that there is no discussion of the overturning circulation on

the shelf in this manuscript and how that might be influenced by the new bathymetry observations.

- *Thank you for the suggestion. As I mentioned above, for this revision, we created the absolute ocean velocity (Fig. 2d-f) from the geostrophic velocity calculation, referenced to the satellite-derived surface ocean velocity, following the same method as Hirano et al. (2021). Since the observations along the upper slope section (Figs. 2a, d) do not follow the isobath, the results may partly include features such as a meandering slope current, but at least the warm water core at ~119.5E corresponds to the downstream of the southward flowing region of the eastern half of the West Sabrina Eddy. The subsequent southward inflow is still observed in the shelf break and inner shelf sections east of 119E (Fig. 2b, c, e, f), where warm water widely spreads near the seafloor. In other words, it is an important observational fact that at least the offshore eddies contribute to the poleward warm-water transport across the shelf break, followed by the continuous southward warm water inflow to the inner shelf. (Line: 97-114)*

Furthermore, we have enhanced simulation results along the ice-front section (Fig.8) and in the TIS cavity (ice-pump overturning circulation, Fig.9). The enhanced observations and simulations significantly improve the picture of the warm water circulation all the way from the offshore into the TIS cavity through the deep channels. (Line: 220-243)

Line 127: “This suggests that TTR1 is the main conduit by which ocean heat is delivered for TIS melting.” Without showing the velocity fields, I am not sure how the authors reach this conclusion — this can not be deduced by only looking at the temperatures. Both the circulation and heat content contribute to the total heat flux.

- *Based on your comment, we have added simulated data along the ice-front sections of temperature, salinity, and ocean velocity (inflow/outflow) (Fig. 8) to complement the lack of ocean velocity observations at the TIS front. We established that the most significant mCDW inflow, marked by the thicker mCDW layer along with the bottom-intensified velocity of 4–8 cm s⁻¹, occurs in the bottom layer (deeper than 500–600 m) centered at the eastern side of the TTR1. Then, we could conclude from the observations (Figs. 3, 4, 5) and simulations (Fig. 8) that the TTR1 directly connects the TIS cavity and the Sabrina Depression (Figs. 1, 3) and is the main conduit by which ocean heat is delivered for TIS melting. (Line: 131-135; 226-234)*

Besides, the TTR2 is also a deep trough with a maximum depth of >1000 m but it seems to be surrounded by relatively shallower topography with <500–600 m depth; therefore, its

topographic connection to the Sabrina Depression is undefined from the existing bathymetric data. The shallow topographic context would prevent the inflows of the warmer mCDW present at the deeper layer of the depression into the TTR2, as inferred by Silvano et al. (2019). (Line: 135-142)

Spatiotemporal variability section: There is no discussion or speculation about possible causes for the sub-seasonal variability. Tides are mentioned at one point, but the authors do not provide information about the expected magnitude of the tides in this region.

- *Based on the comment, we have added possible causes for the sub-seasonal variability in addition to tide-related variations (semidiurnal/diurnal to fortnightly): offshore semi-permanent West Sabrina Eddy with characters with a relatively short variability in 2-6 months (Mizobata et al., 2020) and wind/buoyancy forcing. However, we cannot quantitatively evaluate each contribution to the inflowing mCDW variation with the current dataset because of the reason noted above. (Line: 158-170)*

In addition, we have added pressure time series data as tidal variation in the region (see Fig. 6b).

Line 179: “showing a reasonable model representation of ice shelf-ocean interaction in this region.” Is the parameterization for ice-shelf melt in this model tuned to agree with this value?

- *No, we did not perform tuning to match the simulated basal melting to the satellite-based estimate. In this study, we used an ice-ocean parameterization of Hellmer and Ollbers (1989) based on an observation of ice-ocean interaction. In the revised manuscript, we have updated the model simulation using ERA5 forcing. The modeled basal melting amount at the TIS becomes smaller than that included in the previous manuscript (forced with ERA-Interim). However, the model estimate of 47 Gt/yr is not far from the observational estimate, 63 Gt/yr. The applicable sentences have been rephrased. (Line: 209-210).*

Simulation section: Overall, the modeling work is very nice and provides a useful complement to the observations. However, it feels like the analysis here is somewhat superficial because of the length of the manuscript. The only real comparison in the main text are two mean sections of T and S shown in Figure 4b. There are some significant differences between the thickness of the CDW layer in these panels, but this is not

discussed in detail. Can these be attributed to seasonal fluctuations, uncertainty in surface forcing in the model, other?

- *Thank you very much for your encouragement on our modeling component. We consider that there is a novelty in a direct comparison between the in-situ observation and modeling results. The lack and ambiguity of the bathymetry map off the Sabrina Coast, particularly near the TIS, has hampered the previous modeling studies from simulating ocean circulation and the subsequent TIS-ocean interaction. Our simulation with the updated bathymetry for the first time demonstrated the overall ocean circulation from shelf break, continental shelf, to the region in front of/underneath the TIS. (Line: 63-71; 200-243)*

We acknowledge the differences between the observation and model results; the model biases probably come from integrated uncertainties in snapshot-/averaged-fields, uncharted bathymetry, atmospheric forcing, and model configurations (as you pointed out). As mentioned in earlier responses, we are working on a study that describes the model results, in which we focus on seasonal-to-interannual variability in the model. Thus, we consider that the simple comparison between observation and model climatological results is sufficient to support the observational findings in this study.

Figure 4c: Units are not provided for the streamfunction.

- *Thank you for your correction. The figure was removed in this revision. We have carefully checked the details (units, legends, and captions) of all the figures in the revised manuscript.*

REVIEWER COMMENTS

Reviewer #1 (Remarks to the Author):

The authors have considered the suggestions made by both reviewers, and made some good and useful revisions. They have added a substantial amount of new data from the most recent season, which is great. I would like to be more enthusiastic about this paper, but I feel it could still be strengthened to make the most of this great data set. I hope these suggestions are helpful, as I would like to see this work published.

My primary concern of the original paper was that it needed to make clear what was new or surprising, and why it mattered. This was also raised by Reviewer 2. The authors' response states "to date, there has been no integrated description/knowledge of warm-water distribution and circulation from offshore to the TIS front regions in association with realistic bathymetry" which I didn't find very convincing. I am still unclear exactly what insight this paper has added to our knowledge. Figure 10 is lovely, but couldn't this schematic have been produced before any of the work described in this paper, based on pre-existing papers? The paper argues that this paper presents "more detail" but why are these details important (and for whom/what)? Maybe new insights are presented, but I was not sure what they were. If something is a new finding, say so explicitly.

I still find it hard to identify what is new. The abstract, for example, is unchanged. It makes a series of statements with which everyone can agree, I think, but doesn't make clear what specific bit of knowledge was added through the analysis presented in the paper (that merits publication in Nature Comms). Sentences 1-3 are scene setting, that's fine. The 4th sentence might be new results, or might not – can you clarify for your future readers exactly what is the new finding in this sentence? The 5th sentence is surely obvious and could be written about any ice shelf even with no observations? And the final sentence seems a bit of a stretch – what exactly in the results/analysis has "highlighted the glacier's vulnerability" that we did not know before?

It is good that the authors talked more about the circulation in the revised version. It surprises me that there are no velocity data – does the ship not have a shipboard ADCP? Did the moorings that you mention not have velocity? Heat transports would greatly strengthen the paper. In the abstract and again in line 45 you introduce the concept of ocean heat transport as being key, but you never calculate this so it is unclear why you introduce it. You could quantify it in your super model simulation?

In lines 117-120, you talk about there being a cyclonic circulation of mCDW. If this is the first paper to identify this, then say so. You have the data to quantify it more – how many Sv? If it's not new, then shorten the description so that the paper focuses on the things that are new – this will help your readers to grasp your take-home message.

Reviewer 2 made a very valid suggestion that you should do more than just present data (e.g. time series or sections) (their point 3). I wasn't really convinced by your response. The paragraph added about the mooring time series (p8-9) simply describes the variability in the time series. But what is the message, why do we care about this, is any of this new or surprising? Pointing out the surprises would be very helpful.

What would really demonstrate the value of the revised bathymetry, to me, would be a comparison of a numerical simulation with the "old" bathymetry, and a simulation with the new/revised bathymetry. You state (line 65) that the model "reasonably reproduces the ocean circulations and mCDW pathways from the shelf break to the TIS front regions, consistent with the observational findings". This raises two questions in my mind – first, how do you know this, and do you mean that it agrees with the observations of this paper, or of previous work? Second, if you regard the observations as "truth", what are you gaining from the use of the model? What key result comes from the model? The model is great, but you could use it to quantify or shed light on something? Is it the melt rates? If so, sell this new finding better (e.g. before our new bathymetry, models couldn't get the melt rate correct, but now.....?). You state (line 205) that the model representation (of water masses? Fluxes? Or what?) is "substantially improved" with the new

bathymetry, but how do we know this? What was it like before the new bathymetry?

In my original review, I asked "Did you consider using the O18 measurements to calculate the meltwater distribution?" The authors' response is to state that there is a 0.2% increase in meltwater in the outflow compared with the inflow. The distribution is not shown. Since we're not given the numbers, this isn't very useful. And 0.2% increase is surely not significant given the uncertainties in the measurements and in the calculations of meltwater? In addition, I was puzzled by the references given here? The first reference is about the WAP – it is by no means the earliest paper to discuss calculating the freshwater components from O18 using this method, nor is it from this region, so why is this particular paper an appropriate reference?

Line 263, you state that the "observations provide compelling evidence that warm water reaches the TIS to drive rapid basal melt". But did we not know this already from other papers? If this is new, say so.

Lines 267-275 – I cannot see the relevance of these sentences. They talk about other people's results, and are nothing to do with the implications of your own results. You need to say how your new results affects these speculations. Or remove them.

The figures are much better, thank you, though I find some of the fonts extremely small (e.g. contour labels and station numbers in figure 2).

Figure 2, it would be good if you used colorbarfill or similar (as you did in Figure 3) so that the colorbar is consistent with the filled contours.

For all figures, it would be helpful for readers if there was more consistency, e.g. in colorbars, colormaps, colorbar ranges.

For example:

- Figure 2 e does not have a comparable colorbar range as d and f for velocities
- Figure 7 uses different colormaps and colorbar ranges from the other figures of T and S, and Figure 8 is different again. Then Figure 9 uses yet another temperature colorbar.
- Figure 8 c does not use the same velocity colormap or colorbar range as the earlier velocity sections from observations. It is also given in cm/s rather than m/s – units should be consistent throughout.
- Is the colormap used in Figure 8 a,b colour-blind friendly?

Minor points:

Line 94-95, the sentence about glacial erosion seems irrelevant to this paper and should be removed.

Line 285, I would say suggests rather than offers

Karen J. Heywood

Reviewer #2 (Remarks to the Author):

Review of: On-shelf circulation of warm water toward the Totten Ice Shelf, East Antarctica
Authors: Daisuke Hirano, et al.

This manuscript documents new observations collected over the Antarctic continental shelf in the Sabrina Coast region in front of the Totten Ice Shelf. Specifically, multi beam sonar surveys from multiple research cruises (including a cruise from March 2022 that was not included in the original manuscript) provides details of deep glacial troughs that control the circulation around and under the Totten Ice Shelf. These bathymetric observations are coupled with a mooring time series and high-resolution numerical simulations to update the lateral circulation and heat transport pathways over the continental shelf.

I thank the authors for the thorough response to my original review. I appreciate that multiple changes and additions have been implemented in the revised manuscript, which I feel has strengthened the paper. The new observations collected during the March 2022 cruise also strengthen the main messages of the paper. While I feel that there is still scope for a deeper exploration of the physical processes dictating the seasonal to interannual variability in the heat transport, the partitioning of heat between different troughs and the inclusion of a more thorough analysis of the numerical simulations are important stories. If the authors are able to address my remaining comments and suggestions below, then I would recommend acceptance of this manuscript.

Major comments

- The manuscript now includes estimates of the velocity field in Figure 2, which have a strong barotropic component, especially over the shelf. Therefore the velocity field is strongly dependent on the ADCP processing and analysis. This information is currently missing in the main text as well as in the Methods section.

Re-reading the replies, I now realize that the absolute velocity field is derived from estimating surface velocities from remote sensing data. I could not find this stated in the manuscript. If this is indeed the case, it would be very helpful to know what the spatial resolution of this SSH product. In fact, a more complete description of the altimetry product should be included in the Methods section. It appears to originate from Mizobata et al. (2018), but this is not cited in the manuscript.

- The horizontal resolution of the numerical simulation is 3-4 km but the trough widths are only 10-20 km wide (line 88). Therefore, boundary currents will be poorly resolved in the model, if at all. At most there will be only a couple of points representing both inflow and outflows in the troughs. While I appreciate that it will not be possible to run the model at higher resolution for this study, caveats related to this grid size should be discussed. Does the model capture the temporal variability seen in the mooring data?
- The velocity sections in Figure 2 are a nice addition to the paper — is it possible to include data closer to the coast? It is hard to see from Figure 3 exactly how many CTDs profiles there are from the 2022 work, but it appears that a crude section could be made that spans TTR1 and TTR2.
- The heat transport estimates from the model are still not presented in the manuscript. It seems that a simple comparison between the observations and the model is whether the frequency spectra of temperature variations are similar between the two. Then, the model could test whether variations in mCDW temperature or mCDW layer thickness are correlated with changes in cross-shelf heat transport. It seems odd that a paper detailing heat transport towards Totten Ice Shelf does not diagnose heat transport in the model.

Minor comments

Line 31: "The temperature of mCDW reaching TIS varies on tidal-to-interannual timescales, indicating that ocean-driven melt is sensitive to changes in forcing." Again, I thank the authors for digging a bit deeper into the time series analysis in the manuscript. This particular statement in the abstract is not terribly surprising — a more substantial comment about what timescales are dominant (the authors seem to suggest interannual) would make it clearer to the reader the key result(s) of this study.

Line 38: "has also contributed significantly to sea-level rise in recent decades." Quantify significant?

Paragraph starting on line 45: The second and third sentences say the same thing.

Line 88: "Totten Trough 1 (TTR1)" It feel it would be more intuitive to readers if these were

labeled as TT-E and TT-W for east and west. I found I had to keep referring to the map to remember which was trough 1 and trough 2.

Line 109: "a sharp interface in temperature and salinity (hence density)." You should mention here that the water is salinity stratified, so it is mostly the halocline (and not the thermocline) that generates a pycnocline.

Line 137: This is nice — in addition to the weaker exchange between the depression and TTR2, there may also be more recirculation within TTR2 that gives rise to further homogenization of properties.

The paragraph ending on line 163 is a nice addition to the paper!

Line 165: It would be good to define "West Sabrina Eddy" briefly.

Line 166: "Suppose the season is from autumn to winter. In that case . . . deprive the available oceanic heat of the water column." These sentences need to be re-phrased. I think you simply mean that "Buoyancy forcing may also contribute to sub-seasonal variations in autumn to winter, since deepening . . . removes heat from the water column." Although, why does this only have to be during fall/winter months?

Line 169: What does "their" refer to in this sentence?

Line 170: "If we define trends". This also should be re-worded. You do calculate trends — they are just small.

Line 188 and following: Oxygen isotopies are introduced here without really explaining how they are interpreted. You need at least one sentence explaining this proxy so the study is clear to non-experts.

Line 195: "These signals reflect the mixing of glacial meltwater ..." What type of mixing are you envisioning here: small-scale vertical mixing, lateral stirring along isopycnals, instabilities at the ice-shelf face?

Line 200: Remove or re-word "lack" — observations are lacking, not the bathymetric map.

Line 204: I realize that the model is described in Methods, but it is nice to at least state what model is being used in the main text.

Line 209: "The simulated amount of TIS basal melting ..." Is this value tuned in the simulation? Also, the Methods section should provide information on the parameters that were selected to determine melt rates.

Line 271: Remove "Besides,"

Line 340: The model name/type should be included in the Methods section, e.g. MITgcm, ROMS, etc.

Figure 3: Consider using different color scales for bathymetry and temperature.

Figure 4: It would be very nice to see temperature/salinity diagrams for these profiles — not just temperature. Also, I did not understand from the caption what the triangles in panel (b) were supposed to illustrate.

Figure 6: Consider flipping the y-axis in Figure 6 to make the figure more intuitive (colder water on top)? Maybe this would not be intuitive for everyone!

Figure 8: I do not understand what panel (d) is showing (why are there different colors? How are the "areas" defined? Why is the distance not just continuous?). Also, I am not sure where the ice-

shelf front is — does it coincide with the bold red line?

Figure 9: This is really nice! It is interesting that the Eulerian, depth-based overturning is much larger than the overturning in density space — is this a Deacon-cell like effect? One small request is to include the maximum streamfunction value in the caption.

REVIEWER COMMENTS

We thank both reviewers for their insightful and helpful comments, which have helped us to substantially improve the manuscript. We have responded to each of the comments raised by the reviewers, as discussed below. The reviewer comments are shown in bold black font and our response in blue italics. Changes to the manuscript are highlighted in the attached revision of the manuscript.

Reviewer #1 (Remarks to the Author):

The authors have considered the suggestions made by both reviewers, and made some good and useful revisions. They have added a substantial amount of new data from the most recent season, which is great. I would like to be more enthusiastic about this paper, but I feel it could still be strengthened to make the most of this great data set. I hope these suggestions are helpful, as I would like to see this work published.

Dear Prof. Karen Heywood, We thank you for the time reading our revised manuscript and for the helpful comments provided. Your comments were invaluable for further improving our manuscript. We believe we have addressed all your comments/suggestions and substantially improved the paper.

My primary concern of the original paper was that it needed to make clear what was new or surprising, and why it mattered. This was also raised by Reviewer 2. The authors' response states "to date, there has been no integrated description/knowledge of warm-water distribution and circulation from offshore to the TIS front regions in association with realistic bathymetry" which I didn't find very convincing. I am still unclear exactly what insight this paper has added to our knowledge. Figure 10 is lovely, but couldn't this schematic have been produced before any of the work described in this paper, based on pre-existing papers? The paper argues that this paper presents "more detail" but why are these details important (and for whom/what)? Maybe new insights are presented, but I was not sure what they were. If something is a new finding, say so explicitly.

➤ *We have carefully revised the manuscript to address your concern. We have added a section to the Introduction to highlight the novel aspects of our work. (Line: 48-62, 68-79)*

We have also made sure to explicitly highlight new results and their significance throughout the text. Novel aspects of our work include:

- *The first comprehensive hydrographic and bathymetric observations spanning the Totten continental shelf. These observations revealed for the first time the pathway of warm mCDW from the open ocean, across the shelf, and to the front of the ice shelf. The combined hydrographic and bathymetric observations also showed how specific*

bathymetric features control the transport of warm water from the open ocean to the ice shelf cavity.

- The first hydrographic observations (including oxygen isotope measurements) at the western TIS front, confirming that the outflow of glacial meltwater from the TIS cavity is concentrated in the west.
- The first measurements of variability in the properties of warm mCDW reaching the ice front. These observations showed that the temperature of mCDW varies strongly from year-to-year. The moored time series (the first collected near the Totten ice front) helped explain why: variations in mCDW temperature near the ice front are associated with changes in thermocline depth.
- The first numerical simulations of the region with realistic bathymetry and sufficient resolution to represent the circulation, water mass distributions, ocean heat transport, and ocean – ice shelf interaction. The model simulations, carefully validated against the new observations, allowed us to quantify the circulation on the shelf and in the ice shelf cavity for the first time, including ocean heat transport, and confirmed the importance of the Central Totten Trough as the primary conduit for warm mCDW to reach the ice shelf.
- An additional experiment has been added to this revision, in which the deep troughs are filled in. Comparison of the two simulations provides a quantitative demonstration of the critical role of the deep troughs in allowing warm mCDW to reach the ice shelf cavity.

I still find it hard to identify what is new. The abstract, for example, is unchanged. It makes a series of statements with which everyone can agree, I think, but doesn't make clear what specific bit of knowledge was added through the analysis presented in the paper (that merits publication in Nature Comms). Sentences 1-3 are scene setting, that's fine. The 4th sentence might be new results, or might not – can you clarify for your future readers exactly what is the new finding in this sentence? The 5th sentence is surely obvious and could be written about any ice shelf even with no observations? And the final sentence seems a bit of a stretch – what exactly in the results/analysis has “highlighted the glacier's vulnerability” that we did not know before?

- Thank you for the suggestion on the abstract. We have substantially modified the abstract to convey to readers the paper's novelty based on your comment. (Line: 24-34)

It is good that the authors talked more about the circulation in the revised version. It surprises me that there are no velocity data – does the ship not have a shipboard ADCP? Did the moorings that you mention not have velocity? Heat transports would greatly strengthen the paper. In the abstract and again in line 45 you introduce the concept of ocean heat transport as being key, but

you never calculate this so it is unclear why you introduce it. You could quantify it in your super model simulation?

- *Our icebreaker, “Shirase,” has an SADCP. But unfortunately, we found the SADCP data to be of poor quality on this voyage, probably because the vessel was passing through heavy sea ice for most of the time on the continental shelf. No current meters were installed on the mooring system at East Totten Trough.*

Therefore, as you suggested, we evaluated ocean heat transports from the model simulations in this revision. Notably, a new “No Trough Experiment (see below)” allows us to quantitatively discuss the importance of Totten Troughs in ocean heat transport/partitioning to the TIS cavity.

In lines 117-120, you talk about there being a cyclonic circulation of mCDW. If this is the first paper to identify this, then say so. You have the data to quantify it more – how many Sv? If it’s not new, then shorten the description so that the paper focuses on the things that are new – this will help your readers to grasp your take-home message.

- *A previous paper by Silvano et al. (2019) used trajectories of a small number of under-ice Argo floats to infer a cyclonic circulation in the Sabrina Depression, but the actual shape of the depression was unknown at that time. We have revised the text to make it clear that a previous study had suggested the presence of a cyclonic circulation, which our more complete hydrographic and bathymetric observations confirm. (Line: 126-129)*

We now use the model to quantify the strength of the cyclonic circulation around the rim of the depression. (Line: 228-230)

Reviewer 2 made a very valid suggestion that you should do more than just present data (e.g. time series or sections) (their point 3). I wasn’t really convinced by your response. The paragraph added about the mooring time series (p8-9) simply describes the variability in the time series. But what is the message, why do we care about this, is any of this new or surprising? Pointing out the surprises would be very helpful.

- *We have carefully revised the manuscript throughout to ensure the message is clear. We have articulated the new aspects of the work above and in the revised manuscript text. The revised text provides quantitative analysis of the circulation on the shelf, ocean heat transport, basal melt, and the sub-ice shelf circulation. Comparison of the numerical simulations with and without deep troughs provides a quantitative demonstration of the importance of the Central Totten Trough as a pathway for warm mCDW to reach the ice front. We thank both reviewers for helping us strengthen the analysis and clarify the message.*

With regard to the mooring data, we have now moved this to the Supplementary Information (Supplementary Fig. 5). The 3-month duration of the moored time series is useful (e.g., for showing that tides influence the temperature of mCDW at the ice front), but does not address the issue of interannual variability, which makes the dominant contribution to variability of mCDW at the ice front and hence is our focus here. Furthermore, our numerical model does not incorporate a tide scheme, so it does not make sense to compare the short mooring record, in which the variability is dominated by tides, with the numerical model results.

What would really demonstrate the value of the revised bathymetry, to me, would be a comparison of a numerical simulation with the “old” bathymetry, and a simulation with the new/revised bathymetry. You state (line 65) that the model “reasonably reproduces the ocean circulations and mCDW pathways from the shelf break to the TIS front regions, consistent with the observational findings”. This raises two questions in my mind – first, how do you know this, and do you mean that it agrees with the observations of this paper, or of previous work? Second, if you regard the observations as “truth”, what are you gaining from the use of the model? What key result comes from the model? The model is great, but you could use it to quantify or shed light on something? Is it the melt rates? If so, sell this new finding better (e.g. before our new bathymetry, models couldn’t get the melt rate correct, but now... ”?). You state (line 205) that the model representation (of water masses? Fluxes? Or what?) is “substantially improved” with the new bathymetry, but how do we know this? What was it like before the new bathymetry?

- *Thank you for your suggestion. To demonstrate the real value of the new bathymetry from our multibeam sonar survey, we conducted a new experiment, in which the deep troughs near the ice front are artificially filled in (the "No Trough" experiment, Line: 471-476 in Methods).*

By comparing the results between experiments with and without troughs, we quantitatively evaluated the significance of the deep troughs for ocean heat transports into the TIS cavity (Fig. 8). Note that the "No Trough experiment" imitates a numerical simulation with the old bathymetry dataset that you suggested (as mentioned, essential topographic features such as the Totten Troughs and Sabrina Depression are not correctly expressed in any previous bathymetric datasets). The comparison shows clearly that the Central Totten Trough, first mapped by our intensive bathymetric survey, is the most important conduit for ocean heat to reach the TIS cavity and drive basal melt of the ice shelf. (Line: 250-264 & 274-276)

In my original review, I asked "Did you consider using the O18 measurements to calculate the meltwater distribution?" The authors’ response is to state that there is a 0.2% increase in meltwater in the outflow compared with the inflow. The distribution is not shown. Since we’re

not given the numbers, this isn't very useful. And 0.2% increase is surely not significant given the uncertainties in the measurements and in the calculations of meltwater? In addition, I was puzzled by the references given here? The first reference is about the WAP – it is by no means the earliest paper to discuss calculating the freshwater components from O18 using this method, nor is it from this region, so why is this particular paper an appropriate reference?

- *We apologize for the previous insufficient response. We now compare the actual values, rather than referring to their difference, to avoid any ambiguity. In response to your comment, we have also added a new figure (Fig. 5d) and a description regarding the spatial distribution of the glacial meltwater fraction. We also added Price et al. (2008) as a more appropriate reference for East Antarctic ice shelf as well. (Line: 197-200)*

We have also added an explanation of the error evaluation of meltwater fraction estimates caused by the O18 uncertainty in glacial meltwater endmember to the Method section. Although the absolute values in meltwater fraction have substantial uncertainty in WW layer, we believe their spatial difference has far less uncertainty and the western-central TIS contrast is robust. (See Methods for “Estimation of glacial meltwater fraction”, the new section from this revision)

Line 263, you state that the “observations provide compelling evidence that warm water reaches the TIS to drive rapid basal melt”. But did we not know this already from other papers? If this is new, say so.

- *Based on your comment, we have deleted that sentence and modified this part: "Satellite observations show that the Totten Glacier has thinned and the grounding line has retreated in recent decades^{34, 35}. These changes may reflect an increase in ocean heat transport to the TIS cavity, but this hypothesis cannot be tested with observations because oceanographic measurements at the TIS front (ref. 11 and this study) span a short time interval." (Line: 310-313)*

Lines 267-275 – I cannot see the relevance of these sentences. They talk about other people's results, and are nothing to do with the implications of your own results. You need to say how your new results affects these speculations. Or remove them.

- *Thank you for the comment. We have added an explanation clarifying the relevance of our own results with those sentences. (Line: 317-320)*

The figures are much better, thank you, though I find some of the fonts extremely small (e.g. contour labels and station numbers in figure 2).

- *Thank you, too. We have made the font size as large as possible for figures with small fonts.*

Figure 2, it would be good if you used colorbarfill or similar (as you did in Figure 3) so that the colorbar is consistent with the filled contours.

➤ *We have modified Fig. 2 as suggested.*

For all figures, it would be helpful for readers if there was more consistency, e.g. in colorbars, colormaps, colorbar ranges. For example:

- Figure 2 e does not have a comparable colorbar range as d and f for velocities
- Figure 7 uses different colormaps and colorbar ranges from the other figures of T and S, and Figure 8 is different again. Then Figure 9 uses yet another temperature colorbar.
- Figure 8 c does not use the same velocity colormap or colorbar range as the earlier velocity sections from observations. It is also given in cm/s rather than m/s – units should be consistent throughout.
- Is the colormap used in Figure 8 a,b colour-blind friendly?

➤ *Thank you for the comments. We have made colormaps and units consistent as much as possible. We use colormaps of “RdBu” for temperature and ocean velocity, “Spectral” for salinity, and “RdYlBu” for dissolved oxygen (<https://matplotlib.org/stable/tutorials/colors/colormaps.html>).*

We have made colorbars consistent for Figs. 2d-f. For Fig. 2e, the ocean velocity out of range is represented by contours only.

The ranges of colorbar for the observations and simulation are still different because we think we should highlight the values as obviously as possible for each result.

Furthermore, we checked whether the figures (<https://pilestone.com/pages/color-blindness-simulator-1>), and we confirmed they are discernable with this color space.

Minor points:

Line 94-95, the sentence about glacial erosion seems irrelevant to this paper and should be removed.

➤ *As you suggested, we have deleted the sentence from this revision.*

Line 285, I would say suggests rather than offers

➤ *Thank you for the suggestion. We have modified this part: "The large interannual variability observed in the temperature of mCDW reaching the TIS (Fig. 4a) highlights the need for long-*

term monitoring of the oceanic thermal forcing driving basal melt of the TIS." (Line: 332-335)

Karen J. Heywood

Reviewer #2 (Remarks to the Author):

Review of: On-shelf circulation of warm water toward the Totten Ice Shelf, East Antarctica

Authors: Daisuke Hirano, et al.

This manuscript documents new observations collected over the Antarctic continental shelf in the Sabrina Coast region in front of the Totten Ice Shelf. Specifically, multi beam sonar surveys from multiple research cruises (including a cruise from March 2022 that was not included in the original manuscript) provides details of deep glacial troughs that control the circulation around and under the Totten Ice Shelf. These bathymetric observations are coupled with a mooring time series and high-resolution numerical simulations to update the lateral circulation and heat transport pathways over the continental shelf.

I thank the authors for the thorough response to my original review. I appreciate that multiple changes and additions have been implemented in the revised manuscript, which I feel has strengthened the paper. The new observations collected during the March 2022 cruise also strengthen the main messages of the paper. While I feel that there is still scope for a deeper exploration of the physical processes dictating the seasonal to interannual variability in the heat transport, the partitioning of heat between different troughs and the inclusion of a more thorough analysis of the numerical simulations are important stories. If the authors are able to address my remaining comments and suggestions below, then I would recommend acceptance of this manuscript.

➤ *We thank you for the time reading our revised paper and for the helpful comments provided. We also very much appreciate your positive assessment of the previous revision. Your comments were invaluable for further improving our manuscript. We believe we have addressed all your comments/suggestions and substantially improved the paper. In particular, we have estimated the ocean heat transports into the TIS cavity by the simulation and could further highlight through a new additional model experiment the role of regional circulation and bathymetry in regulating ocean heat transports into the TIS cavity for rapid melting in the Totten Glacier.*

Major comments

- The manuscript now includes estimates of the velocity field in Figure 2, which have a strong barotropic component, especially over the shelf. Therefore the velocity field is strongly dependent on the ADCP processing and analysis. This information is currently missing in the main text as well as in the Methods section.

Re-reading the replies, I now realize that the absolute velocity field is derived from estimating surface velocities from remote sensing data. I could not find this stated in the manuscript. If this is indeed the case, it would be very helpful to know what the spatial resolution of this SSH product. In fact, a more complete description of the altimetry product should be included in the Methods section. It appears to originate from Mizobata et al. (2018), but this is not cited in the manuscript.

➤ *Sorry for the confusion. Based on your comment, we have added a new section to the Methods with further information on the SSH products (Mizobata et al., 2020) and now they are used to derive absolute ocean velocity from SSH and hydrographic data (according to Hirano et al., 2021). (See Methods for “Calculation of absolute ocean velocity from hydrographic and satellite-derived Dynamic Ocean Topography (DOT) data”, the new section from this revision)*

• **The horizontal resolution of the numerical simulation is 3-4 km but the trough widths are only 10-20 km wide (line 88). Therefore, boundary currents will be poorly resolved in the model, if at all. At most there will be only a couple of points representing both inflow and outflows in the troughs. While I appreciate that it will not be possible to run the model at higher resolution for this study, caveats related to this grid size should be discussed. Does the model capture the temporal variability seen in the mooring data?**

➤ *Based on the suggestion, we have added a brief comment regarding the model grid size and highlighted what the model can or cannot represent. (Line: 218-221)*

Our numerical model does not incorporate a tide scheme, so it does not make sense to compare the short mooring record, in which the variability is dominated by tides, with the numerical model results. As our focus here is on the interannual variability that dominates the changes observed in mCDW from repeat hydrography, we have moved the mooring data to the Supplementary Information (Supplementary Fig. 5).

We have added one figure presenting results from the new numerical model experiment without troughs to highlight the Central Trough's significance in ocean heat transports into the TIS cavity (Fig. 8).

• **The velocity sections in Figure 2 are a nice addition to the paper — is it possible to include data closer to the coast? It is hard to see from Figure 3 exactly how many CTDs profiles there are from the 2022 work, but it appears that a crude section could be made that spans TTR1 and TTR2.**

➤ *Thank you for the comment. Unfortunately, it is impossible to accurately estimate SSH near the*

coast due to the influence of sea ice and landfast ice. To obtain SSH, there must be open water (leads and cracks in sea ice areas) within the 15 km footprint of the CryoSat-2/SIRAL satellite radar altimeter (the area the sensor is looking at, i.e., the field of view). In the coastal area of this research area, there is no open water surface due to the constant presence of sea ice and landfast ice, so even if an icebreaker could make hydrographic observations, we cannot calculate the absolute velocity because satellite data is not available.

Additionally, there are no hydrographic profiles across the Totten Troughs, so it is impossible to create an observation-based ocean velocity section along the ice front. Accordingly, in the last revision, we added a simulated ocean velocity section along the ice front (Fig. 7). Further, in this revision, we have estimated ocean heat transport through the Totten Troughs and added a quantitative reference to the ocean heat transport to the Sabrina Coast ice shelves (Fig. 8, see also later)

• **The heat transport estimates from the model are still not presented in the manuscript. It seems that a simple comparison between the observations and the model is whether the frequency spectra of temperature variations are similar between the two. Then, the model could test whether variations in mCDW temperature or mCDW layer thickness are correlated with changes in cross-shelf heat transport. It seems odd that a paper detailing heat transport towards Totten Ice Shelf does not diagnose heat transport in the model.**

➤ *As mentioned above, our numerical model does not incorporate a tidal scheme, so a direct comparison with the short mooring time series, which is dominated by tidal signals, does not make sense. So, we moved the mooring time series to the Supplement Information as support data (Supplementary Fig. 5).*

Motivated by this comment, we conducted an additional experiment ("No Trough" experiment Line: 471-476 in Methods) in this revision. In the new experiment, the simulation was conducted by filling the deep layers below 500 m along the Sabrina Coast (i.e., filling the troughs along the ice fronts).

By comparing the results between experiments with and without troughs, we quantitatively evaluated the significance of the deep troughs for ocean heat transports into the TIS cavity (Fig. 8). The comparison clearly shows that the Central Totten Trough, first mapped by our intensive bathymetric survey, is the most important conduit of ocean heat transport along the Sabrina Coast for driving the rapid TIS basal melt.

Minor comments

Line 31: “The temperature of mCDW reaching TIS varies on tidal-to-interannual timescales, indicating that ocean-driven melt is sensitive to changes in forcing.” Again, I thank the authors for digging a bit deeper into the time series analysis in the manuscript. This particular statement in the abstract is not terribly surprising — a more substantial comment about what timescales are dominant (the authors seem to suggest interannual) would make it clearer to the reader the key result(s) of this study.

- *We agree with your comment. We have modified this part to focus more on interannual variability. (Line: 30-31)*

Line 38: “has also contributed significantly to sea-level rise in recent decades.” Quantify significant?

- *We have modified this part slightly: “has also been a major contributor to sea-level rise in recent decades.” (Line: 37-38)*

Paragraph starting on line 45: The second and third sentences say the same thing.

- *Thank you for pointing this out. We have deleted the repetition here. (Line: 44)*

Line 88: “Totten Trough 1 (TTR1)” It feel it would be more intuitive to readers if these were labeled as TT-E and TT-W for east and west. I found I had to keep referring to the map to remember which was trough 1 and trough 2.

- *Thank you for the suggestion. We have named Totten Trough 1 (TTR1) as Central Totten Trough (C-TT) and Totten Trough 2 (TTR2) as East Totten Trough (E-TT). (Lines: 99-100 & 103-104)*

Note: *The east trough is located on the eastern ice front (so it would be OK to refer to this trough as suggested), but the western of the two troughs is located in the central ice front (so it could be confusing to refer to this trough as “TT-W”). To avoid any confusion, we refer to the two troughs as Central Totten Trough and East Totten Trough.*

Line 109: “a sharp interface in temperature and salinity (hence density).” You should mention here that the water is salinity stratified, so it is mostly the halocline (and not the thermocline) that generates a pycnocline.

- *We have added a note that the seawater density in the Antarctic coastal region is determined mainly by salinity. (Line: 119-120)*

Line 137: This is nice — in addition to the weaker exchange between the depression and TTR2,

there may also be more recirculation within TTR2 that gives rise to further homogenization of properties.

- *Thank you. We agree with your suggestion of recirculation and a resulting homogenization of the warm water properties within the E-TT. However, we do not have enough evidence, so we kept it here as the original (but added a brief explanation based on your comment). (Line: 144-145)*

The paragraph ending on line 163 is a nice addition to the paper!

- *Thanks! Due to other changes, we have substantially revised the content and structure of this section, but the same statement still ends this section. (Line: 157-184)*

Line 165: It would be good to define “West Sabrina Eddy” briefly.

- *To focus more on the discussion of interannual variation, we have removed this content regarding offshore eddies in this section.*

Line 166: “Suppose the season is from autumn to winter. In that case . . . deprive the available oceanic heat of the water column.” These sentences needs to re-phrased. I think you simply mean that “Buoyancy forcing may also contribute to sub-seasonal variations in autumn to winter, since deepening . . . removes heat from the water column.” Although, why does this only have to be during fall/winter months?

- *We agree with your comment, but to focus more on the discussion of interannual variation, we have removed this content as well in this revision.*

Line 169: What does “their” refer to in this sentence?

- *Same as above, we have deleted the relevant sentence in this revision.*

Line 170”. “If we define trends”. This also should be re-worded. You do calculate trends — they are just small.

- *We agree with you. We have modified the relevant part: “much smaller variability (± 0.1 °C).” (Line: 170-172)*

Line 188 and following: Oxygen isotopies are introduced here without really explaining how they are interpreted. You need at least one sentence explaining this proxy so the study is clear to non-experts.

- *We have briefly explained the oxygen isotope ratio and why we use this proxy. (Line: 192-195)*

Line 195: “These signals reflect the mixing of glacial meltwater ...” What type of mixing are you envisioning here: small-scale vertical mixing, lateral stirring along isopycnals, instabilities at the ice-shelf face?

- *Although it could not be determined, we have also mentioned some possibilities, such as boundary mixing and entrainment by buoyant meltwater plumes. (Line: 205-208)*

Line 200: Remove or re-word “lack” — observations are lacking, not the bathymetric map.

- *Thank you for the correction. We have modified it here. (Line: 211-215)*

Line 204: I realize that the model is described in Methods, but it is nice to at least state what model is being used in the main text.

Line 340: The model name/type should be included in the Methods section, e.g. MITgcm, ROMS, etc.

- *Thank you very much for your advice. In the revised manuscript, we have added the model's name (COCO) in the main text and Methods section. (Line: 215-217 & 429)*

It should be noted that COCO originally stood for CCSR Ocean COMponent model, but the research center, CCSR, was merged into AORI, the University of Tokyo. Now we have used COCO as a proper noun for pointing out the ice-ocean model.

Line 209: “The simulated amount of TIS basal melting ...” Is this value tuned in the simulation? Also, the Methods section should provide information on the parameters that were selected to determine melt rates.

- *We used an observation-based parameter of the thermal and salinity exchange velocities ($\gamma_t = 1.0 \times 10^{-4} \text{ m s}^{-1}$ and $\gamma_s = 5.05 \times 10^{-7} \text{ m s}^{-1}$, Hellmer and Olbers (1989)) in this study. (Line: 456-458)*

We didn't perform turning on the parameter, and the rate and amount of the model's ice-shelf basal melting are estimated from ice-ocean interaction with the parameterization in the model configuration.

Line 271: Remove “Besides,”

- *We have modified it here. (Line: 320)*

Figure 3: Consider using different color scales for bathymetry and temperature.

- *Thank you for the suggestion. We have modified this figure with each colormap for bathymetry and water temperature. In addition, we originally displayed bathymetric data from the multi-beam survey in the broad map (Fig. 3b, now it is Fig. 3a), but we have removed the bathymetry*

from the broad map for more facilitating visualization.

Figure 4: It would be very nice to see temperature/salinity diagrams for these profiles — not just temperature. Also, I did not understand from the caption what the triangles in panel (b) were supposed to illustrate.

- Since the T-S diagram does not allow us to highlight the interannual variations in mCDW that we focus on here (all plots for the mCDW layer are almost along the same line), we have added a vertical salinity profile as panel b of Figure 4.

Note that we have moved the original Fig.4b to Supplementary Fig. 5a. The modified caption of Supplementary Fig. 5 is, “Right-pointing triangles indicate the depths of the moored temperature time series plotted in panel (d), using the same colors (showing which depths the sensors measured).” (see Supplementary Fig. 5)

Figure 6: Consider flipping the y-axis in Figure 6 to make the figure more intuitive (colder water on top)? Maybe this would not be intuitive for everyone!

- Thanks for the suggestion. We have tried to flip the y-axis of the time series (see below). However, the flipped version would make warm and cold cycles of water temperatures misleading (upward means cooler in this case). So, we think it should be kept original.

The flipped version of Supplementary Fig. 5d.

Figure 8: I do not understand what panel (d) is showing (why are there different colors? How are the “areas” defined? Why is the distance not just continuous?). Also, I am not sure where the ice-shelf front is — does it coincide with the bold red line?

- We have modified the caption (the current Figure 7d). Line sections are required to integrate volume transport in lateral and vertical directions to calculate the overturning ocean circulation. Since the model's horizontal resolution is about 3-4 km, we categorized the model grids with a 5 km interval based on distance from the ice-front line, and the boundaries between the different categorized grids were used for the line sections. Colors in the panel show the category and the boundaries. We have added orange lines showing the ice front on the figure.

Figure 9: This is really nice! It is interesting that the Eulerian, depth-based overturning is much larger than the overturning in density space — is this a Deacon-cell like effect? One small request is to include the maximum streamfunction value in the caption.

- Thank you so much. The targeted ice shelf is connected to the Sabrina Depression with several deep troughs of different depths. Panel (a) in the depth-distance domain is intuitive, but a combination of the realistic topography and the ocean circulation makes interpretation difficult. We consider that the overturning circulation in the density-distance domain is more suitable for understanding ocean thermohaline circulation within the cavity in the realistic topography configuration. Following your suggestions, we have added numbers of the streamfunction at the ice front and the position of the maximum absolute value in the density-distance domain.

REVIEWER COMMENTS

Reviewer #1 (Remarks to the Author):

Thank you for responding thoroughly and carefully to my suggestions and those of the other reviewer. I particularly appreciate the additional model run and the revised figures, thank you. The novelty of the paper is now much clearer for readers. I am delighted to recommend publication.

Reviewer #2 (Remarks to the Author):

Review of: On-shelf circulation of warm water toward the Totten Ice Shelf, East Antarctica
Authors: Daisuke Hirano, et al.

This manuscript documents new observations collected over the Antarctic continental shelf in the Sabrina Coast region in front of the Totten Ice Shelf. Specifically, multi beam sonar surveys from multiple research cruises, including a cruise from March 2022, which provides details of deep glacial troughs that control the circulation around and under the Totten Ice Shelf. These bathymetric observations are coupled with a mooring time series and high-resolution numerical simulations to update the lateral circulation and heat transport pathways over the continental shelf. In the latest revised version the authors have focused more on inter-annual variability in mCDW properties, as opposed to the tidal variations revealed in the mooring time series, and they have added a new numerical simulation that highlights the impact of the troughs on heat transport towards the Totten Ice Shelf.

I again appreciate the thorough replies to my second round of reviews. The manuscript continues to improve. A few questions about the analysis still remain, which are documented below. However, it feels like a straightforward and important addition to the manuscript would be to inform the reader whether the model reproduces the observed interannual variability in mCDW properties — this would strengthen the paper significantly. It would be great to address the points below as the authors are preparing a final version of the manuscript and the editor decides on suitability for publication.

Major comments

- Throughout the manuscript (and in the reviewer replies) there are inconsistencies in whether the changes in temperature of the mCDW or the thickness of the mCDW layer are more likely to impact ice shelf melt rates. From Figure 4, it looks like the thermocline depth only varies by ~100 m, whereas temperature varies by over a degree. Therefore temperature differences are likely as important, if not more important, than variations in thermocline/pycnocline depth. It is worth noting in the manuscript that this is a very different system from the Amundsen Sea, where mCDW temperatures have been largely unchanged over the past decade and a half, but the thermocline depth varies by 300-400 m.
- Line 56-58: “to determine the pathways and processes regulating the transport of warm water ... to assess the variability in the temperature of mCDW.” This is a nice addition to the manuscript that clearly lays out the goals of the study. The revised manuscript makes a much clearer statement that the inter-annual variability in mCDW characteristics are more important than the tidal fluctuations from the mooring data. However, there is no discussion in the current version of the manuscript about what processes give rise to this interannual variability, other than it is not due to Ekman transport at the shelf break. Even some speculation about how these changes might be occurring would strengthen the paper. Also, I realize that the model does not have tides, so comparison with the mooring data is not possible, but does the model reproduce the different mCDW regimes seen in the observations? Again, these observations are very exciting and worthy of publication — however, for a Nat. Comm. publication, I would expect a discussion of mechanism as well as reporting of the observations.

- I appreciate the additional information provided about the SSH product that provides the reference values for the estimates of total geostrophic velocity and transport. There are some issues with this approach that need to be discussed in the manuscript. First, as mentioned previously, the flow is almost purely barotropic, so the velocities are strongly influenced by the choice of reference velocity. There is a mismatch between the SSH reference, which is a time-averaged velocity over a period of almost a decade and the synoptic sections that were collected during the cruises. During the time of the cruise, the barotropic velocity might be considerably different from what is inferred from the SSH product — this is why it is much better to reference to ADCP data, if available. I also appreciate that ADCP data is not available. The authors should consider whether they want to continue to use the SSH velocity product — in which case it needs to be made clear in the manuscript that this is a climatological velocity field and not directly related to the observation period. Alternatively, the authors could just show the baroclinic component of the flow, which will be more closely linked to the ice-shelf overturning circulation (e.g. Wahlin et al. Nature 2020)

Minor comments

- As the authors are preparing a final version of this manuscript I would urge them to be more quantitative into the abstract. Rather than just saying that temperature varies, provide a range; rather than just saying that bathymetry changes the heat transport in the simulations, give a percentage of how much based on the new numerical simulation.
- Line 37: “a major contributor to sea level rise” I recognize the change from “significant” to “major” but I was hoping that you would be able to provide a quantitative value for the reader.
- Line 43: “and a net mass loss of ...” over which period?
- Somewhere in the manuscript there should be a comment that the velocities in the upper slope section (Figure 2d) are all offshore. How does the heat access the continental shelf in this case? Standing eddies are mentioned multiple times, but then I would expect to see a pattern of onshore and offshore flow.
- Line 182-183: “we do not find clear evidence of a relationship between zonal wind along the Sabrina Coast and WW/mCDW interface variations. This suggests that wind-driven Ekman upwelling/downwelling is not the dominant process driving thermocline depth variations.” Be careful here that you are not mixing Ekman transport (dependent on wind stress) and Ekman pumping (dependent on wind stress curl). A recent paper by Kim et al. (Kim et al JGR, 2021) shows that wind stress curl, integrated over the entire continental shelf, does a good job of reproducing changes in the WW/mCDW interface in front of Dotson ice shelf.
- Thank you for including the salinity profiles in Figure 4. It is fascinating the variations in salinity (stratification) look quite different from what is observed in the Amundsen Sea.
- Line 221: “special” should be “spatial”
- Line 258: I could not find where the acronym mMUIS was defined.
- Figure 8: The x-axis should be labeled.
- Line 291: “Because the depth of the thermocline is similar to the depth of the shallowest sills, heaving of the sharp thermocline between cold WW and warm mCDW drives large variations in the temperature of mCDW reaching the cavity.” What causes this heaving?
- Line 326: “ In particular, our observations from the TIS front show that changes in depth of the thermocline drive variability in thermal forcing on broad range of timescales (tidal to interannual).” This statement seems to be different from the abstract which talks about changes in the temperature of mCDW, rather than the thermocline depth.

- Line 430: "commponnt" should be "component"
- Line 470: Since you have run the model through the entire period where there are observations, it would be really interesting to know if the model is able to reproduce the observed variability in mCDW thickness and temperature!

REVIEWER COMMENTS

Reviewer #1 (Remarks to the Author):

Thank you for responding thoroughly and carefully to my suggestions and those of the other reviewer. I particularly appreciate the additional model run and the revised figures, thank you. The novelty of the paper is now much clearer for readers. I am delighted to recommend publication.

Our sincere gratitude to Prof. Karen Heywood:

We heartily thank you for your constructive and helpful comments up to this point, which were invaluable for improving our manuscript.

Reviewer #2 (Remarks to the Author):

Review of: On-shelf circulation of warm water toward the Totten Ice Shelf, East Antarctica

Authors: Daisuke Hirano, et al.

This manuscript documents new observations collected over the Antarctic continental shelf in the Sabrina Coast region in front of the Totten Ice Shelf. Specifically, multi beam sonar surveys from multiple research cruises, including a cruise from March 2022, which provides details of deep glacial troughs that control the circulation around and under the Totten Ice Shelf. These bathymetric observations are coupled with a mooring time series and high-resolution numerical simulations to update the lateral circulation and heat transport pathways over the continental shelf. In the latest revised version the authors have focused more on inter-annual variability in mCDW properties, as opposed to the tidal variations revealed in the mooring time series, and they have added a new numerical simulation that highlights the impact of the troughs on heat transport towards the Totten Ice Shelf.

I again appreciate the thorough replies to my second round of reviews. The manuscript continues to improve. A few questions about the analysis still remain, which are documented below. However, it feels like a straightforward and important addition to the manuscript would be to inform the reader whether the model reproduces the observed interannual variability in mCDW properties — this would strengthen the paper significantly. It would be great to address the points below as the authors are preparing a final version of the manuscript and the editor decides on suitability for publication.

We thank Reviewer #2 for the helpful comments, which have helped us to improve the manuscript further. We have responded to each of the comments raised by the reviewer, as discussed below. The reviewer comments are shown in bold black font, and our response is in blue italics. Changes to the manuscript are highlighted in the attached revision of the manuscript.

Major comments

- Throughout the manuscript (and in the reviewer replies) there are inconsistencies in whether the changes in temperature of the mCDW or the thickness of the mCDW layer are more likely to impact ice shelf melt rates. From Figure 4, it looks like the thermocline depth only varies by ~100 m, whereas temperature varies by over a degree. Therefore temperature differences are likely as important, if not more important, than variations in thermocline/pycnocline depth. It is worth noting in the manuscript that this is a very different

system from the Amundsen Sea, where mCDW temperatures have been largely unchanged over the past decade and a half, but the thermocline depth varies by 300-400 m.

• Line 56-58: “to determine the pathways and processes regulating the transport of warm water ... to assess the variability in the temperature of mCDW.” This is a nice addition to the manuscript that clearly lays out the goals of the study. The revised manuscript makes a much clearer statement that the inter-annual variability in mCDW characteristics are more important than the tidal fluctuations from the mooring data. However, there is no discussion in the current version of the manuscript about what processes give rise to this interannual variability, other than it is not due to Ekman transport at the shelf break. Even some speculation about how these changes might be occurring would strengthen the paper. Also, I realize that the model does not have tides, so comparison with the mooring data is not possible, but does the model reproduce the different mCDW regimes seen in the observations? Again, these observations are very exciting and worthy of publication — however, for a Nat. Comm. Publication, I would expect a discussion of mechanism as well as reporting of the observations.

- *Thank you for your comments, which have helped us to further deepen and improve the discussion. Specifically, we have added a new discussion on the possible processes driving the variability of temperature and thermocline depth on the Antarctic continental shelf (Line: 292-328), including a comparison of the depth range of the thermocline on the Amundsen Sea continental shelf in West Antarctica (Line: 300-304). In addition, we have provided additional relevant comments in some places (Line: 137-139; 150-151).*
- *Thank you also for the comment on the numerical model. We confirmed that our numerical model reproduces the different mCDW regimes observed at the TIS front to some extent (see the figure below). Due to space and figure constraints, we decided to focus on the mean fields (climatology) for the simulation results in this paper, and it sufficiently supports our observational findings. However, as you pointed out, we also consider that the interannual variation of the cryosphere-ocean system is important and should be addressed using the model results. We have prepared a separate numerical simulation paper entitled “Modeling seasonal-to-decadal ocean-cryosphere interactions along the Sabrina Coast, East Antarctica” by Kusahara, Hirano, et al., and it is now ready for submission to a journal (The Cryosphere) after the acceptance of this paper.*

Figure: A simulated different regime of inflowing mCDW property at the TIS front (shown by blue circle), corresponding to observations shown in Fig. 4. Instead of Mar. 2022 in Fig. 4, the model results for Dec. 2021 (the end of the model simulation) were used.

- I appreciate the additional information provided about the SSH product that provides the reference values for the estimates of total geostrophic velocity and transport. There are some issues with this approach that need to be discussed in the manuscript. First, as mentioned previously, the flow is almost purely barotropic, so the velocities are strongly influenced by the choice of reference velocity. There is a mismatch between the SSH reference, which is a time-averaged velocity over a period of almost a decade and the synoptic sections that were collected during the cruises. During the time of the cruise, the barotropic velocity might be considerably different from what is inferred from the SSH product — this is why it is much better to reference to ADCP data, if available. I also appreciate that ADCP data is not available. The authors should consider whether they want to continue to use the SSH velocity product — in which case it needs to be made clear in the manuscript that this is a climatological velocity field and not directly related to the observation period. Alternatively, the authors could just show the baroclinic component of the flow, which will be more closely linked to the ice-shelf overturning circulation (e.g. Wahlin et al. Nature 2020)

- *Apologies for not being entirely clear in the Methods. We use ocean surface velocity derived from monthly, not climatological, SSH data, using the closest month to the observation period, as a reference velocity for calculating the absolute ocean velocity for Fig. 2.*

Specifically, we used the monthly mean SSH in February 2020 because the observation data used for Fig. 2 were obtained from February to early March. Note that the choice of monthly SSH in March 2020 shows almost no change in the results. We have added one sentence to Methods for clarity. (Line: 430-432)

Minor comments

- **As the authors are preparing a final version of this manuscript I would urge them to be more quantitative into the abstract. Rather than just saying that temperature varies, provide a range; rather than just saying that bathymetry changes the heat transport in the simulations, give a percentage of how much based on the new numerical simulation.**

- *Thank you for your suggestion. We have now stated the range of the observed mCDW temperature reaching the TIS cavity in the Abstract (Line: 30-31). On the other hand, since the experimental setup of the simulation with no troughs is somewhat arbitrary, the quantitative impact of the bathymetry on ocean heat transport may change depending on the experimental setup (i.e., how the troughs are filled). Therefore, we have not provided a quantitative value for the change in heat transport between the two simulations in the Abstract. However, readers can easily find the difference in ocean heat transport between the experiments in Fig. 8.*

- **Line 37: “a major contributor to sea level rise” I recognize the change from “significant” to “major” but I was hoping that you would be able to provide a quantitative value for the reader.**

- *Based on your comment, we have modified this sentence as “While the contribution to global sea-level rise from West Antarctica has received the most attention (6.9 ± 0.6 mm for the period 1979-2017, dominated by the Amundsen and Bellingshausen Sea sectors), East Antarctica has also been a major contributor (4.4 ± 0.9 mm over the same period)¹. (Line: 36-38)*

- **Line 43: “and a net mass loss of ...” over which period?**

- *We have provided the period for which the net mass loss was estimated. (Line: 44)*

- Somewhere in the manuscript there should be a comment that the velocities in the upper slope section (Figure 2d) are all offshore. How does the heat access the continental shelf in this case? Standing eddies are mentioned multiple times, but then I would expect to see a pattern of onshore and offshore flow.

- *Hirano et al. (2021) showed cross-slope mCDW intrusion onto the continental shelf with meridional sections in this region, and we can see also from Supplementary Fig.2 that the Totten continental shelf is connected to the offshore via the standing eddy off the shelf; however, the hydrographic sections (Fig. 2) are not ideally suited for illustrating this connection, as you pointed out. Therefore, we have revised the first paragraph of the section “Pathways of Warm Water from the Shelf Break to the TIS Front.” (Line: 110-116)*

- Line 182-183: “we do not find clear evidence of a relationship between zonal wind along the Sabrina Coast and WW/mCDW interface variations. This suggests that wind-driven Ekman upwelling/downwelling is not the dominant process driving thermocline depth variations.” Be careful here that you are not mixing Ekman transport (dependent on wind stress) and Ekman pumping (dependent on wind stress curl). A recent paper by Kim et al. (Kim et al JGR, 2021) shows that wind stress curl, integrated over the entire continental shelf, does a good job of reproducing changes in the WW/mCDW interface in front of Dotson ice shelf.

- *Thank you for your information. Kim et al. (2021), with long-term oceanographic surveys from 2007 to 2018, found that the interannual mCDW variability is coherent with local Ekman pumping integrated along the Dotson-Getz Trough. In contrast, our observations themselves are too sparse to allow a statistically robust analysis of the causal link between oceanic interannual variability and forcing. Therefore, it would be best to consider Kim’s approach when we obtain long-term observation data in the Totten region. Instead, we have cited Kim et al. (2021) in the new discussion (Line: 317) and included relevant comments (Line: 326-328; 356-358; 359-360).*

- Thank you for including the salinity profiles in Figure 4. It is fascinating the variations in salinity (stratification) look quite different from what is observed in the Amundsen Sea.

- *As in the earlier response to the first two major comments, we have included the comparison of the depth range of the thermocline on the Amundsen Sea continental shelf in West Antarctica (Line: 300-304).*

- Line 221: “special” should be “spatial”

- *Thank you. We have corrected it. (Line: 224)*

- **Line 258: I could not find where the acronym mMUIS was defined.**

➤ *Thank you for pointing this out. We have modified this sentence as “Warm water also reaches the cavities beneath the nearby eastern TIS (eTIS) and western Moscow University Ice Shelf (wMUIS), but the heat transport is relatively small (118 GW at eTIS, 300 GW at wMUIS).” (Line: 260-261)*

- **Figure 8: The x-axis should be labeled.**

➤ *We have added “Month” on the x-axis. (Revised Figure 8)*

- **Line 291: “Because the depth of the thermocline is similar to the depth of the shallowest sills, heaving of the sharp thermocline between cold WW and warm mCDW drives large variations in the temperature of mCDW reaching the cavity.” What causes this heaving?**

➤ *As in the earlier response to the first two major comments, we have added a new discussion on the possible processes driving the variability of temperature and thermocline depth on the Antarctic continental shelf (Line: 292-328), including the change of this sentence (Line: 304-306).*

Note that variations in temperature and variations in thermocline depth are related, as discussed in the paper. The observations suggest that variations in temperature at the ice front (i.e., of most relevance to basal melt) are strongly linked to changes in depth of the thermocline relative to the depth of the shallowest sills.

- **Line 326: “ In particular, our observations from the TIS front show that changes in depth of the thermocline drive variability in thermal forcing on broad range of timescales (tidal to interannual).” This statement seems to be different from the abstract which talks about changes in the temperature of mCDW, rather than the thermocline depth.**

➤ *Following the abovementioned changes, we have slightly weakened this sentence's claim (Line: 351-354).*

- **Line 430: “commponnt” should be “component”**

➤ *Thank you for pointing this out. We have corrected it (Line: 458).*

- **Line 470: Since you have run the model through the entire period where there are observations, it would be really interesting to know if the model is able to reproduce the observed variability in mCDW thickness and temperature!**

➤ *Thank you very much for your encouragement. As in the earlier response, we will submit another modeling paper about the seasonal-to-decadal variability (Kusahara, Hirano et al.,*

to be submitted to *The Cryosphere*). We have examined the temporal variability of the mCDW in the modeling paper in detail.